# Task-dependent optimal representations for cerebellar learning

Marjorie Xie[1], Samuel P Muscinelli[1], Kameron Decker Harris[2], Ashok Litwin-Kumar[1]*

[1]Zuckerman Mind Brain Behavior Institute, Columbia University, New York, United States; [2]Department of Computer Science, Western Washington University, Bellingham, United States

**Abstract** The cerebellar granule cell layer has inspired numerous theoretical models of neural representations that support learned behaviors, beginning with the work of Marr and Albus. In these models, granule cells form a sparse, combinatorial encoding of diverse sensorimotor inputs. Such sparse representations are optimal for learning to discriminate random stimuli. However, recent observations of dense, low-dimensional activity across granule cells have called into question the role of sparse coding in these neurons. Here, we generalize theories of cerebellar learning to determine the optimal granule cell representation for tasks beyond random stimulus discrimination, including continuous input-output transformations as required for smooth motor control. We show that for such tasks, the optimal granule cell representation is substantially denser than predicted by classical theories. Our results provide a general theory of learning in cerebellum-like systems and suggest that optimal cerebellar representations are task-dependent.

## Editor's evaluation

Models of cerebellar function and the coding of inputs in the cerebellum often assume that random stimuli are a reasonable stand-in for real stimuli. However, the important contribution of this paper is that conclusions about optimality and sparseness in these models do not generalize to potentially more realistic sets of stimuli, for example, those drawn from a low-dimensional manifold. The mathematical analyses in the paper are convincing and possible limitations, including the abstraction from biological details, are well discussed.

**\*For correspondence:**
a.litwin-kumar@columbia.edu

**Competing interest:** The authors declare that no competing interests exist.

## Introduction

A striking property of cerebellar anatomy is the vast expansion in number of granule cells compared to the mossy fibers that innervate them (*Eccles et al., 1967*). This anatomical feature has led to the proposal that the function of the granule cell layer is to produce a high-dimensional representation of lower-dimensional mossy fiber activity (*Marr, 1969*; *Albus, 1971*; *Cayco-Gajic and Silver, 2019*). In such theories, granule cells integrate information from multiple mossy fibers and respond in a nonlinear manner to different input combinations. Detailed theoretical analysis has argued that anatomical parameters such as the ratio of granule cells to mossy fibers (*Babadi and Sompolinsky, 2014*), the number of inputs received by individual granule cells (*Litwin-Kumar et al., 2017*; *Cayco-Gajic et al., 2017*), and the distribution of granule-cell-to-Purkinje-cell synaptic weights *Brunel et al., 2004* have quantitative values that maximize the dimension of the granule cell representation and learning capacity. Sparse activity, which increases dimension, is also cited as a key property of this representation (*Marr, 1969*; *Albus, 1971*; *Babadi and Sompolinsky, 2014*; but see *Spanne and Jörntell, 2015*). Sparsity affects both learning speed (*Cayco-Gajic et al., 2017*) and generalization, the

ability to predict correct labels for previously unseen inputs (*Barak et al., 2013*; *Babadi and Sompolinsky, 2014*; *Litwin-Kumar et al., 2017*).

Theories that study the effects of dimension on learning typically focus on the ability of a system to perform categorization tasks with random, high-dimensional inputs (*Barak et al., 2013*; *Babadi and Sompolinsky, 2014*; *Litwin-Kumar et al., 2017*; *Cayco-Gajic et al., 2017*). In this case, increasing the dimension of the granule cell representation increases the number of inputs that can be discriminated. However, cerebellar circuits participate in diverse behaviors, including dexterous movements, inter-limb coordination, the formation of internal models, and cognitive behaviors (*Ito and Itō, 1984*; *Wolpert et al., 1998*; *Strick et al., 2009*). Cerebellum-like circuits, such as the insect mushroom body and the electrosensory system of electric fish, support other functions such as associative learning (*Modi et al., 2020*) and the cancellation of self-generated sensory signals (*Kennedy et al., 2014*), respectively. This diversity raises the question of whether learning high-dimensional categorization tasks is a sufficient framework for probing the function of granule cells and their analogs.

Several recent studies have reported dense activity in cerebellar granule cells in response to sensory stimulation or during motor control tasks (*Jörntell and Ekerot, 2006*; *Knogler et al., 2017*; *Wagner et al., 2017*; *Giovannucci et al., 2017*; *Badura and De Zeeuw, 2017*; *Wagner et al., 2019*), at odds with classical theories (*Marr, 1969*; *Albus, 1971*). Moreover, there is evidence that granule cell firing rates differ across cerebellar regions (*Heath et al., 2014*; *Witter and De Zeeuw, 2015*). In contrast to this reported dense activity in cerebellar granule cells, odor responses in Kenyon cells, the analogs of granule cells in the *Drosophila* mushroom body, are sparse, with 5–10% of neurons responding to odor stimulation (*Turner et al., 2008*; *Honegger et al., 2011*; *Lin et al., 2014*).

We propose that these differences can be explained by the capacity of representations with different levels of sparsity to support learning of different tasks. We show that the optimal level of sparsity depends on the structure of the input-output relationship of a task. When learning input-output mappings for motor control tasks, the optimal granule cell representation is much denser than predicted by previous analyses. To explain this result, we develop an analytic theory that predicts the performance of cerebellum-like circuits for arbitrary learning tasks. The theory describes how properties of cerebellar architecture and activity control these networks' inductive bias: the tendency of a network toward learning particular types of input-output mappings (*Sollich, 1998*; *Jacot et al., 2018*; *Bordelon et al., 2020*; *Canatar et al., 2021b*; *Simon et al., 2021*). The theory shows that inductive bias, rather than the dimension of the representation alone, is necessary to explain learning performance across tasks. It also suggests that cerebellar regions specialized for different functions may adjust the sparsity of their granule cell representations depending on the task.

## Results

In our model, a granule cell layer of $M$ neurons receives connections from a random subset of $N$ mossy fiber inputs. Because $M \gg N$ in the cerebellar cortex and cerebellum-like structures (approximately $M = 200,000$ and $N = 7,000$ for the neurons presynaptic to a single Purkinje cell in the cat brain; *Eccles*

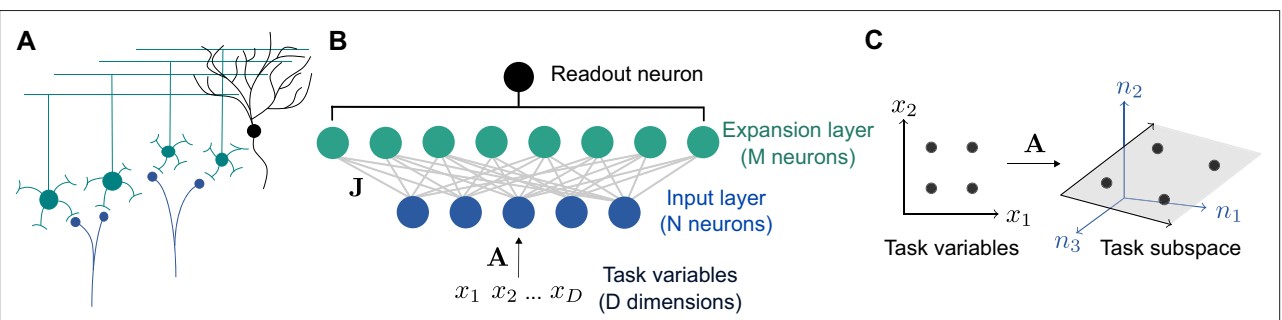

**Figure 1.** Schematic of cerebellar cortex model. (**A**) Mossy fiber inputs (blue) project to granule cells (green), which send parallel fibers that contact a Purkinje cell (black). (**B**) Diagram of neural network model. $D$ task variables are embedded, via a linear transformation **A**, in the activity of $N$ input layer neurons. Connections from the input layer to the expansion layer are described by a synaptic weight matrix **J**. (**C**) Illustration of task subspace. Points **x** in a $D$-dimensional space of task variables are embedded in a $D$-dimensional subspace of the $N$-dimensional input layer activity **n** ($D$=2, $N$=3 illustrated).

*et al., 1967*), we refer to the granule cell layer as the expansion layer and the mossy fiber layer as the input layer (*Figure 1A and B*).

A typical assumption in computational theories of the cerebellar cortex is that inputs are randomly distributed in a high-dimensional space (*Marr, 1969*; *Albus, 1971*; *Brunel et al., 2004*; *Babadi and Sompolinsky, 2014*; *Billings et al., 2014*; *Litwin-Kumar et al., 2017*). While this may be a reasonable simplification in some cases, many tasks, including cerebellum-dependent tasks, are likely best-described as being encoded by a low-dimensional set of variables. For example, the cerebellum is often hypothesized to learn a forward model for motor control (*Wolpert et al., 1998*), which uses sensory input and motor efference to predict an effector's future state. Mossy fiber activity recorded in monkeys correlates with position and velocity during natural movement (*van Kan et al., 1993*). Sources of motor efference copies include motor cortex, whose population activity lies on a low-dimensional manifold (*Wagner et al., 2019*; *Huang et al., 2013*; *Churchland et al., 2010*; *Yu et al., 2009*). We begin by modeling the low dimensionality of inputs and later consider more specific tasks.

We therefore assume that the inputs to our model lie on a $D$-dimensional subspace embedded in the $N$-dimensional input space, where $D$ is typically much smaller than $N$ (*Figure 1B*). We refer to this subspace as the 'task subspace' (*Figure 1C*). A location in this subspace is described by a $D$ dimensional vector $\mathbf{x}$, while the corresponding input layer activity is given by $\mathbf{n} = \mathbf{Ax}$, with $\mathbf{A}$ an $N \times D$ matrix describing the embedding of the task variables in the input layer. An $M \times D$ effective weight matrix $\mathbf{J}^{\text{eff}} = \mathbf{JAJ}^{\text{eff}}$, which describes the selectivity of expansion layer neurons to task variables, is determined by $\mathbf{A}$ and the $M \times N$ input-to-expansion-layer synaptic weight matrix $\mathbf{J}$. The activity of neurons in the expansion layer is given by:

$$\mathbf{h} = \phi(\mathbf{J}^{\text{eff}}\mathbf{x} - \theta), \tag{1}$$

where $\phi$ is a rectified linear activation function $\phi(u) = \max(u, 0)$ applied element-wise. Our results also hold for other threshold-polynomial activation functions. The scalar threshold $\theta$ is shared across neurons and controls the coding level, which we denote by $f$, defined as the average fraction of neurons in the expansion layer that are active. We show results for $f < 0.5$, since extremely dense codes are rarely observed in experiments (*Olshausen and Field, 2004*; see Discussion). For analytical tractability, we begin with the case where the entries of $\mathbf{J}^{\text{eff}}$ are independent Gaussian random variables, as in previous theories (*Rigotti et al., 2013*; *Barak et al., 2013*; *Babadi and Sompolinsky, 2014*). This holds when the columns of $\mathbf{A}$ are orthonormal (ensuring that the embedding of the task variables in the input layer preserves their geometry) and the entries of $\mathbf{J}$ are independent and Gaussian. Later, we will show that networks with more realistic connectivity behave similarly to this case.

## Optimal coding level is task-dependent

In our model, a learning task is defined by a mapping from task variables $\mathbf{x}$ to an output $f(\mathbf{x})$, representing a target change in activity of a readout neuron, for example a Purkinje cell. The limited scope of this definition implies our results should not strongly depend on the influence of the readout neuron on downstream circuits. The readout adjusts its incoming synaptic weights from the expansion layer to better approximate this target output. For example, for an associative learning task in which sensory stimuli are classified into categories such as appetitive or aversive, the task may be represented as a mapping from inputs to two discrete firing rates corresponding to the readout's response to stimuli of each category (*Figure 2A*). In contrast, for a forward model, in which the consequences of motor commands are computed using a model of movement dynamics, an input encoding the current sensorimotor state is mapped to a continuous output representing the readout neuron's tuning to a predicted sensory variable (*Figure 2B*).

To examine how properties of the expansion layer representation influence learning performance across tasks, we designed two families of tasks: one modeling categorization of random stimuli, which is often used to study the performance of expanded neural representations (*Rigotti et al., 2013*; *Barak et al., 2013*; *Babadi and Sompolinsky, 2014*; *Litwin-Kumar et al., 2017*; *Cayco-Gajic et al., 2017*), and the other modeling learning of a continuously varying output. The former we refer to as a 'random categorization task' and is parameterized by the number of input pattern-to-category associations $P$ learned during training (*Figure 2C*). During the training phase, the network learns to associate random input patterns $\mathbf{x}^{\mu} \in \mathbb{R}^D$ for $\mu = 1, \ldots, P$ with random binary categories $y^{\mu} = \pm 1$. The elements of $\mathbf{x}^{\mu}$ are drawn i.i.d. from a normal distribution with mean 0 and variance $1/D$. We refer

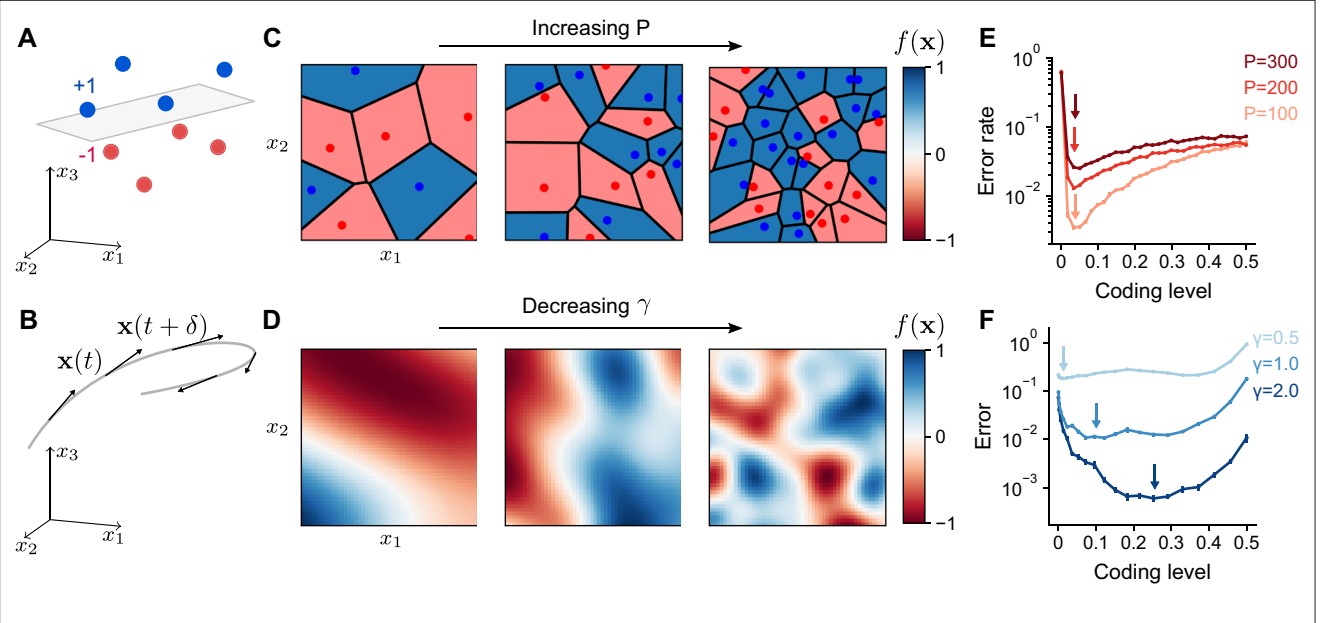

**Figure 2.** Optimal coding level depends on task. (**A**) A random categorization task in which inputs are mapped to one of two categories (+1 or –1). Gray plane denotes the decision boundary of a linear classifier separating the two categories. (**B**) A motor control task in which inputs are the sensorimotor states $\mathbf{x}(t)$ of an effector which change continuously along a trajectory (gray) and outputs are components of predicted future states $\mathbf{x}(t+\delta)$. (**C**) Schematic of random categorization tasks with $P$ input-category associations. The value of the target function $f(\mathbf{x})$ (color) is a function of two task variables $x_1$ and $x_2$. (**D**) Schematic of tasks involving learning a continuously varying Gaussian process target parameterized by a length scale $\gamma$. (**E**) Error rate as a function of coding level for networks trained to perform random categorization tasks similar to (**C**). Arrows mark estimated locations of minima. (**F**) Error as a function of coding level for networks trained to fit target functions sampled from Gaussian processes. Curves represent different values of the length scale parameter $\gamma$. Standard error of the mean is computed across 20 realizations of network weights and sampled target functions in (**E**) and 200 in (**F**).

The online version of this article includes the following figure supplement(s) for figure 2:

**Figure supplement 1.** Sparse coding levels are sufficient for random categorization tasks irrespective of number of samples, noise level, and dimension.

**Figure supplement 2.** Task-dependence of optimal coding level is consistent across activation functions.

**Figure supplement 3.** Task-dependence of optimal coding level is consistent across input dimensions.

**Figure supplement 4.** Error as a function of coding level across different values of $P$ and $\gamma$.

to $\mathbf{x}^\mu$ as 'training patterns'. To assess the network's generalization performance, it is presented with 'test patterns' generated by adding noise (parameterized by a noise magnitude $\epsilon$; see Methods) to the training patterns. Tasks with continuous outputs (**Figure 2D**) are parameterized by a length scale that determines how quickly the output changes as a function of the input (specifically, input-output functions are drawn from a Gaussian process with length scale $\gamma$ for variations in $f(\mathbf{x})$ as a function of $\mathbf{x}$; see Methods). In this case, both training and test patterns are drawn uniformly on the unit sphere. Later, we will also consider tasks implemented by specific cerebellum-like systems. See **Table 1** for a summary of parameters throughout this study.

We trained the readout to approximate the target output for training patterns and generalize to unseen test patterns. The network's prediction is $\hat{f}(\mathbf{x}) = \mathbf{w} \cdot \mathbf{h}(\mathbf{x})$ for tasks with continuous outputs, or $\hat{f}(\mathbf{x}) = \mathrm{sign}(\mathbf{w} \cdot \mathbf{h}(\mathbf{x}))$ for categorization tasks, where $\mathbf{w}$ are the synaptic weights of the readout from the expansion layer. These weights were set using least squares regression. Performance was measured as the fraction of incorrect predictions for categorization tasks, or relative mean squared error for tasks with continuous targets: $\mathrm{Error} = \frac{\mathbb{E}[(f(\mathbf{x}) - \hat{f}(\mathbf{x}))^2]}{\mathbb{E}[f(\mathbf{x})^2]}$, where the expectation is across test patterns.

We began by examining the dependence of learning performance on the coding level of the expansion layer. For random categorization tasks, performance is maximized at low coding levels (**Figure 2E**), consistent with previous results (**Barak et al., 2013**; **Babadi and Sompolinsky, 2014**). The optimal coding level remains below 0.1 in the model, regardless of the number of associations $P$, the

**Table 1.** Summary of simulation parameters.

$M$: number of expansion layer neurons. $N$: number of input layer neurons. $K$: number of connections from input layer to a single expansion layer neuron. $S$: total number of connections from input to expansion layer. $f$: expansion layer coding level. $D$: number of task variables. $P$: number of training patterns. $\gamma$: Gaussian process length scale. $\epsilon$: magnitude of noise for random categorization tasks. We do not report $N$ and $K$ for simulations in which $\mathbf{J}^{\text{eff}}$ contains Gaussian i.i.d. elements as results do not depend on these parameters in this case.

| Figure panel | Network parameters | Task parameters |
|---|---|---|
| *Figure 2E* | $M = 10{,}000$ | $D = 50, P = 1{,}000, \epsilon = 0.1$ |
| *Figures 2F, 4G and 5B* (full) | $M = 200{,}000$ | $D = 3, P = 30$ |
| *Figure 5B and E* | $M = 200{,}000, N = 7{,}000, K = 4$ | $D = 3, P = 30$ |
| *Figure 6A* | $S = MK = 10{,}000, N = 100, f = 0.3$ | $D = 3, P = 200$ |
| *Figure 6B* | $N = 700, K = 4, f = 0.3$ | $D = 3, P = 200$ |
| *Figure 6C* | $M = 5{,}000, f = 0.3$ | $D = 3, P = 100, \gamma = 1$ |
| *Figure 6D* | $M = 1{,}000$ | $D = 3, P = 50$ |
| *Figure 7A* | $M = 20{,}000$ | $D = 6, P = 100$; see Methods |
| *Figure 7B* | $M = 10{,}000, N = 50, K = 7$ | $D = 50, P = 100, \epsilon = 0.1$ |
| *Figure 7C* | $M = 20{,}000, N = 206, 1 \le K \le 3$ | see Methods |
| *Figure 7D* | $M = 20{,}000, N = K = 24$ | $D = 1, P = 30$; see Methods |
| *Figure 2—figure supplement 1* | $M = 10{,}000$ | See Figure |
| *Figure 2—figure supplement 2* | $M = 20{,}000$ | $D = 3, P = 30$ |
| *Figure 2—figure supplement 3* | $M = 20{,}000$ | $D = 3, P = 30$ |
| *Figure 2—figure supplement 4* | $M = 20{,}000$ | $D = 3$ |
| *Figure 7—figure supplement 1* | $M = 20{,}000$ | $D = 3, P = 200$ |
| *Figure 7—figure supplement 2* | $M = 10{,}000, f = 0.3$ | $D = 3, P = 30, \gamma = 1$ |

level of input noise, and the dimension $D$ (*Figure 2—figure supplement 1*). For continuously varying outputs, the dependence is qualitatively different (*Figure 2F*). The optimal coding level depends strongly on the length scale, with learning performance for slowly varying functions optimized at much higher coding levels than quickly varying functions. This dependence holds for different choices of threshold-nonlinear functions (*Figure 2—figure supplement 2*) or input dimension (*Figure 2—figure supplement 3*) and is most pronounced when the number of training patterns is limited (*Figure 2—figure supplement 4*). Our observations are at odds with previous theories of the role of sparse granule cell representations (*Marr, 1969*; *Albus, 1971*; *Babadi and Sompolinsky, 2014*; *Billings et al., 2014*) and show that sparse activity does not always optimize performance for this broader set of tasks.

## Geometry of the expansion layer representation

To determine how the optimal coding level depends on the task, we begin by quantifying how the expansion layer transforms the geometry of the task subspace. Later we will address how this transformation affects the ability of the network to learn a target. For ease of analysis, we will assume for now that inputs are normalized, $\|\mathbf{x}\| = 1$, so that they lie on the surface of a sphere in $D$ dimensions. The set of neurons in the expansion layer activated by an input $\mathbf{x}$ are those neurons $i$ for which the alignment of their effective weights with the input, $\mathbf{J}_i^{\text{eff}} \cdot \mathbf{x}$, exceeds the activation threshold $\theta$ (*Equation 1*; *Figure 3A*). Increasing $\theta$ reduces the size of this set of neurons and hence reduces the coding level.

Different inputs activate different sets of neurons, and more similar inputs activate sets with greater overlap. As the coding level is reduced, this overlap is also reduced (*Figure 3B*). In fact, this reduction

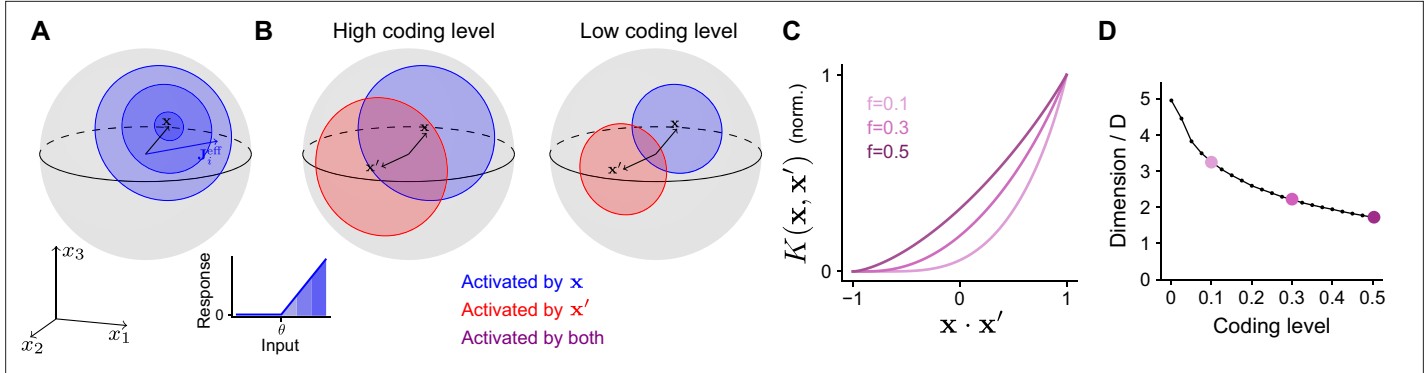

**Figure 3.** Effect of coding level on the expansion layer representation. (**A**) Effect of activation threshold on coding level. A point on the surface of the sphere represents a neuron with effective weights $\mathbf{J}_i^{\text{eff}}$. Blue region represents the set of neurons activated by $\mathbf{x}$, i.e., neurons whose input exceeds the activation threshold $\theta$ (inset). Darker regions denote higher activation. (**B**) Effect of coding level on the overlap between population responses to different inputs. Blue and red regions represent the neurons activated by $\mathbf{x}$ and $\mathbf{x}'$, respectively. Overlap (purple) represents the set of neurons activated by both stimuli. High coding level leads to more active neurons and greater overlap. (**C**) Kernel $K(\mathbf{x}, \mathbf{x}')$ for networks with rectified linear activation functions (**Equation 1**), normalized so that fully overlapping representations have an overlap of 1, plotted as a function of overlap in the space of task variables. The vertical axis corresponds to the ratio of the area of the purple region to the area of the red or blue regions in (**B**). Each curve corresponds to the kernel of an infinite-width network with a different coding level $f$. (**D**) Dimension of the expansion layer representation as a function of coding level for a network with $M = 10,000$ and $D = 3$.

in overlap is greater than the reduction in number of neurons that respond to either of the individual inputs, reflecting the fact that representations with low coding levels perform 'pattern separation' (**Figure 3B**, compare purple and red or blue regions).

This effect is summarized by the 'kernel' of the network (**Schölkopf and Smola, 2002**; **Rahimi and Recht, 2007**), which measures overlap of representations in the expansion layer as a function of the task variables:

$$K(\mathbf{x}, \mathbf{x}') = \frac{1}{M}\mathbf{h}(\mathbf{x}) \cdot \mathbf{h}(\mathbf{x}'). \tag{2}$$

**Equations 1 and 2** show that the threshold $\theta$, which determines the coding level, influences the kernel through its effect on the expansion layer activity $\mathbf{h}(\mathbf{x})$. When inputs are normalized and the effective weights are Gaussian, we compute a semi-analytic expression for the kernel of the expansion layer in the limit of a large expansion ($M \to \infty$; see Appendix). In this case, the kernel depends only on the overlap of the task variables, $K(\mathbf{x}, \mathbf{x}') = K(\mathbf{x} \cdot \mathbf{x}')$. Plotting the kernel for different choices of coding level demonstrates that representations with lower coding levels exhibit greater pattern separation (**Figure 3C**; **Babadi and Sompolinsky, 2014**). This is consistent with the observation that decreasing the coding level increases the dimension of the representation (**Figure 3D**).

## Frequency decomposition of kernel and task explains optimal coding level

We now relate the geometry of the expansion layer representation to performance across the tasks we have considered. Previous studies focused on high-dimensional, random categorization tasks in which inputs belong to a small number of well-separated clusters whose centers are random uncorrelated patterns. Generalization is assessed by adding noise to previously observed training patterns (**Babadi and Sompolinsky, 2014**; **Litwin-Kumar et al., 2017**; **Figure 4A**). In this case, performance depends only on overlaps at two spatial scales: the overlap between training patterns belonging to different clusters, which is small, and the overlap between training and test patterns belonging to the same cluster, which is large (**Figure 4B**). For such tasks, the kernel evaluated near these two values—specifically, the behavior of $K(\mathbf{x} \cdot \mathbf{x}')$ near $\mathbf{x} \cdot \mathbf{x}' = 0$ and $\mathbf{x} \cdot \mathbf{x}' = 1 - \Delta$, where $\Delta$ is a measure of within-cluster noise—fully determines generalization performance (**Figure 4C**; see Appendix). Sparse expansion layer representations reduce the overlap of patterns belonging to different clusters, increasing dimension and generalization performance (**Figure 3D**, **Figure 2E**).

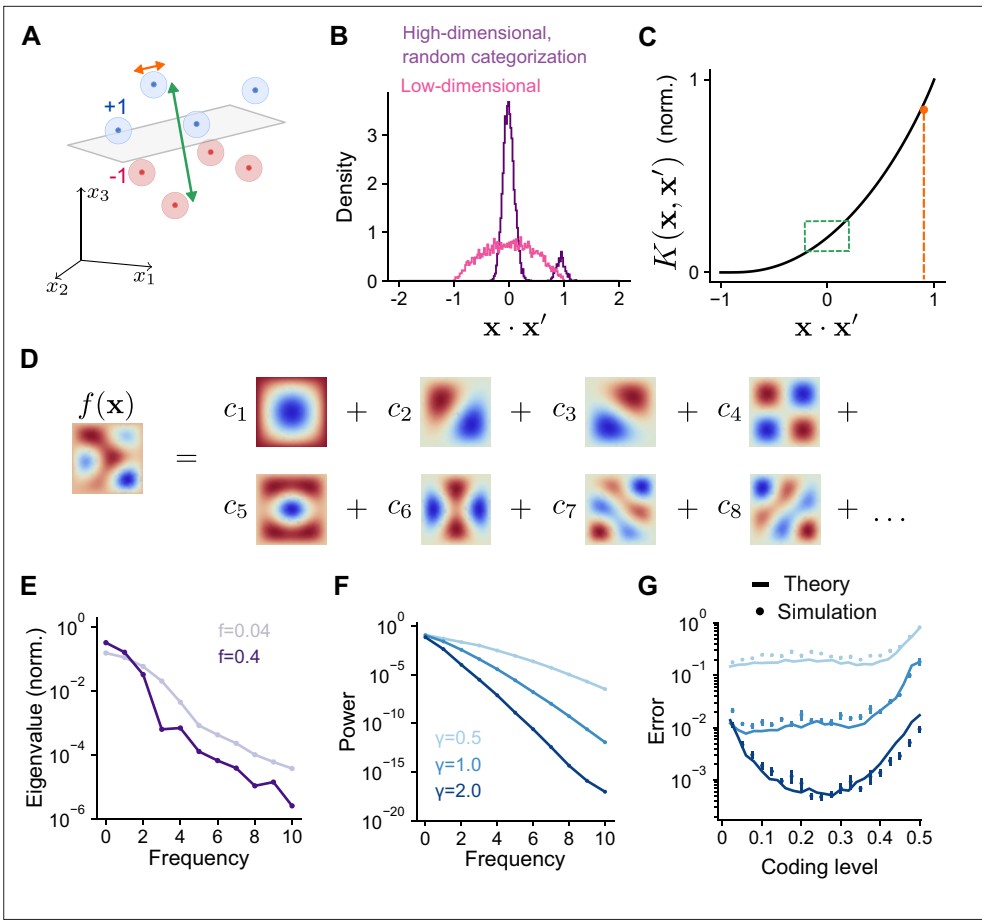

**Figure 4.** Frequency decomposition of network and target function. (**A**) Geometry of high-dimensional categorization tasks where input patterns are drawn from random, noisy clusters (light regions). Performance depends on overlaps between training patterns from different clusters (green) and on overlaps between training and test patterns from the same cluster (orange). (**B**) Distribution of overlaps of training and test patterns in the space of task variables for a high-dimensional task ($D = 200$) with random, clustered inputs as in (**A**) and a low-dimensional task ($D = 5$) with inputs drawn uniformly on a sphere. (**C**) Overlaps in (**A**) mapped onto the kernel function. Overlaps between training patterns from different clusters are small (green). Overlaps between training and test patterns from the same cluster are large (orange). (**D**) Schematic illustration of basis function decomposition, for eigenfunctions on a square domain. (**E**) Kernel eigenvalues (normalized by the sum of eigenvalues across modes) as a function of frequency for networks with different coding levels. (**F**) Power $c_\alpha^2$ as a function of frequency for Gaussian process target functions. Curves represent different values of $\gamma$, the length scale of the Gaussian process. Power is averaged over 20 realizations of target functions. (**G**) Generalization error predicted using kernel eigenvalues (**E**) and target function decomposition (**F**) for the three target function classes shown in (**F**). Standard error of the mean is computed across 100 realizations of network weights and target functions.

The online version of this article includes the following figure supplement(s) for figure 4:

**Figure supplement 1.** Error as a function of coding level for learning pure-frequency spherical harmonic functions.

**Figure supplement 2.** Frequency content of categorization tasks.

We study tasks where training patterns used for learning and test patterns used to assess generalization are both drawn according to a distribution over a low-dimensional space of task variables. While the mean overlap between pairs of random patterns remains zero regardless of dimension, fluctuations around the mean increase when the space is low dimensional, leading to a broader distribution of overlaps (*Figure 4B*). In this case, generalization performance depends on values of the kernel function evaluated across this entire range of overlaps. Methods from the theory of kernel regression (*Sollich, 1998*; *Jacot et al., 2018*; *Bordelon et al., 2020*; *Canatar et al., 2021b*; *Simon et al., 2021*)

capture these effects by quantifying a network's performance on a learning task through a decomposition of the target function into a set of basis functions (*Figure 4D*). Performance is assessed by summing the contribution of each mode in this decomposition to generalization error.

The decomposition expresses the kernel as a sum of eigenfunctions weighted by eigenvalues, $K(\mathbf{x}, \mathbf{x}') = \sum_\alpha \lambda_\alpha \psi_\alpha(\mathbf{x}) \psi_\alpha(\mathbf{x}')$. The eigenfunctions are determined by the network architecture and the distribution of inputs. As we show below, the eigenvalues $\lambda_\alpha$ determine the ease with which each corresponding eigenfunction $\psi_\alpha(\mathbf{x})$—one element of the basis function decomposition—is learned by the network. Under our present assumptions of Gaussian effective weights and uniformly distributed, normalized input patterns, the eigenfunctions are the spherical harmonic functions. These functions are ordered by increasing frequency, with higher frequencies corresponding to functions that vary more quickly as a function of the task variables. Spherical harmonics are defined for any input dimension; for example, in two dimensions they are the Fourier modes. We find that coding level substantially changes the frequency dependence of the eigenvalues associated with these eigenfunctions (*Figure 4E*). Higher coding levels increase the relative magnitude of the low frequency eigenvalues compared to high-frequency eigenvalues. As we will show, this results in a different inductive bias for networks with different coding levels.

To calculate learning performance for an arbitrary task, we decompose the target function in the same basis as that of the kernel:

$$f(\mathbf{x}) = \sum_\alpha c_\alpha \psi_\alpha(\mathbf{x}) \tag{3}$$

The coefficient $c_\alpha$ quantifies the weight of mode $\alpha$ in the decomposition. For the Gaussian process targets, we have considered, increasing length scale corresponds to a greater relative contribution of low versus high frequency modes (*Figure 4F*). Using these coefficients and the eigenvalues (*Figure 4E*), we obtain an analytical prediction of the mean-squared generalization error ('Error') for learning any given task (*Figure 4G*; see Methods):

$$\text{Error} = C_1 \sum_\alpha \left( \frac{c_\alpha}{C_2 + \lambda_\alpha} \right)^2, \tag{4}$$

where $C_1$ and $C_2$ do not depend on $\alpha$ (*Canatar et al., 2021b*; *Simon et al., 2021*; see Methods). *Equation 4* illustrates that for equal values of $c_\alpha$, modes with greater $\lambda_\alpha$ contribute less to the generalization error.

Our theory reveals that the optima observed in *Figure 2F* are a consequence of the difference in eigenvalues of networks with different coding levels. This reflects an inductive bias (*Sollich, 1998*; *Jacot et al., 2018*; *Bordelon et al., 2020*; *Canatar et al., 2021b*; *Simon et al., 2021*) of networks with low and high coding levels toward the learning of high and low frequency functions, respectively (*Figure 4E*, *Figure 4—figure supplement 1*). Thus, the coding level's effect on a network's inductive bias, rather than dimension alone, determines learning performance. Previous studies that focused only on random categorization tasks did not observe this dependence, since errors in such tasks are dominated by the learning of high frequency components, for which sparse activity is optimal (*Figure 4—figure supplement 2*).

## Performance of sparsely connected expansions

To simplify our analysis, we have so far assumed full connectivity between input and expansion layers without a constraint on excitatory or inhibitory synaptic weights. In particular, we have assumed that the effective weight matrix $\mathbf{J}^{\text{eff}}$ contains independent Gaussian entries (*Figure 5A*, top). However, synaptic connections between mossy fibers and granule cells are sparse and excitatory (*Sargent et al., 2005*), with a typical in-degree of $K = 4$ mossy fibers per granule cell (*Figure 5A*, bottom). We therefore analyzed the performance of model networks with more realistic connectivity. Surprisingly, when $\mathbf{J}$ is sparse and nonnegative, both overall generalization performance and the task-dependence of optimal coding level remain unchanged (*Figure 5B*).

To understand this result, we examined how $\mathbf{J}$ and $\mathbf{A}$ shape the statistics of the effective weights onto the expansion layer neurons $\mathbf{J}^{\text{eff}}$. A desirable property of the expansion layer representation is that these effective weights sample the space of task variables uniformly (*Figure 3A*), increasing the heterogeneity of tuning of expansion layer neurons (*Litwin-Kumar et al., 2017*). This occurs when $\mathbf{J}^{\text{eff}}$

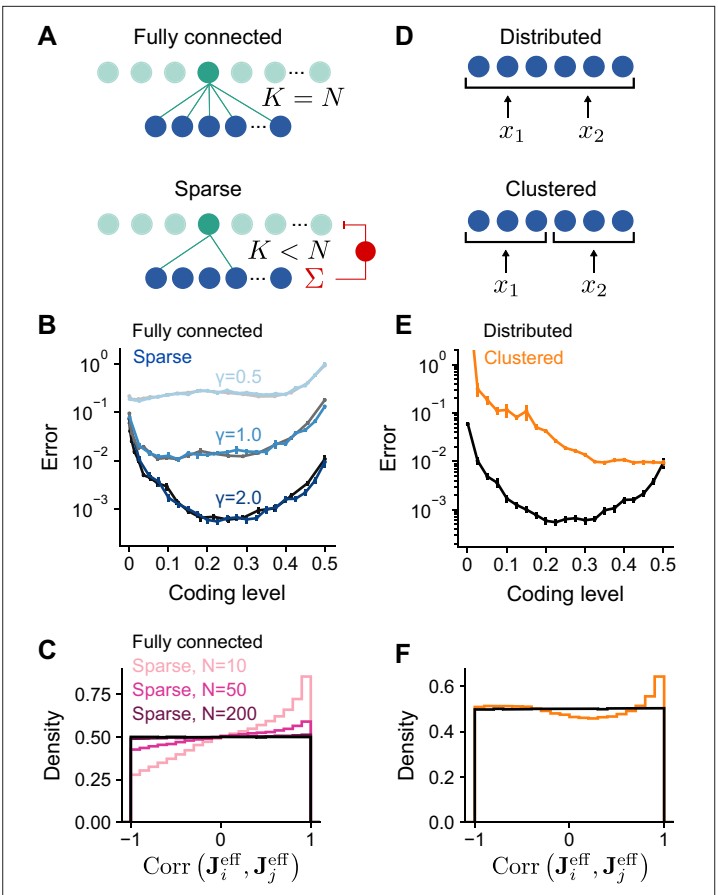

**Figure 5.** Performance of networks with sparse connectivity. (**A**) Top: Fully connected network. Bottom: Sparsely connected network with in-degree $K < N$ and excitatory weights with global inhibition onto expansion layer neurons. (**B**) Error as a function of coding level for fully connected Gaussian weights (gray curves) and sparse excitatory weights (blue curves). Target functions are drawn from Gaussian processes with different values of length scale $\gamma$ as in *Figure 2*. (**C**) Distributions of synaptic weight correlations $\mathbf{Corr}\left(\mathbf{J}_i^{\mathrm{eff}}, \mathbf{J}_j^{\mathrm{eff}}\right)$, where $\mathbf{J}_i^{\mathrm{eff}}$ is the $i$th row of $\mathbf{J}^{\mathrm{eff}}$, for pairs of expansion layer neurons in networks with different numbers of input layer neurons $N$ (colors) when $K = 4$ and $D = 3$. Black distribution corresponds to fully connected networks with Gaussian weights. We note that when $D = 3$, the distribution of correlations for random Gaussian weight vectors is uniform on $[-1, 1]$ as shown (for higher dimensions the distribution has a peak at 0). (**D**) Schematic of the selectivity of input layer neurons to task variables in distributed and clustered representations. (**E**) Error as a function of coding level for networks with distributed (black, same as in **B**) and clustered (orange) representations. (**F**) Distributions of $\mathbf{Corr}\left(\mathbf{J}_i^{\mathrm{eff}}, \mathbf{J}_j^{\mathrm{eff}}\right)$ for pairs of expansion layer neurons in networks with distributed and clustered input representations when $K = 4$, $D = 3$, and $N = 1,000$. Standard error of the mean was computed across 200 realizations in (**B**) and 100 in (**E**), orange curve.

is a matrix of independent random Gaussian entries. If the columns of $\mathbf{A}$ are orthornormal and $\mathbf{J}$ is fully-connected with independent Gaussian entries, $\mathbf{J}^{\mathrm{eff}}$ has this uniform sampling property.

However, when $\mathbf{J}$ is sparse and nonnegative, expansion layer neurons that share connections from the same input layer neurons receive correlated input currents. When $N$ is small and $\mathbf{A}$ is random, fluctuations in $\mathbf{A}$ lead to biases in the input layer's sampling of task variables which are inherited by the expansion layer. We quantify this by computing the distribution of correlations between the effective weights for pairs of expansion layer neurons, $\mathbf{Corr}\left(\mathbf{J}_i^{\mathrm{eff}}, \mathbf{J}_j^{\mathrm{eff}}\right)$. This distribution indeed deviates from uniform sampling when $N$ is small (*Figure 5C*). However, even when $N$ is moderately large (but much less than $M$), only small deviations from uniform sampling of task variables occur for low dimensional tasks as long as $D < K \ll N$ (see Appendix). In contrast, for high-dimensional tasks ($D \sim N$), $K \ll D$ is sufficient, in agreement with previous findings (*Litwin-Kumar et al., 2017*). For realistic cerebellar parameters ($N = 7,000$ and $K = 4$), the distribution is almost indistinguishable from that corresponding

to uniform sampling (*Figure 5C*), consistent with the similar learning performance of these two cases (*Figure 5B*).

In the above analysis, an important assumption is that **A** is dense and random, so that the input layer forms a distributed representation in which each input layer neuron responds to a random combination of task variables (*Figure 5D*, top). If, on the other hand, the input layer forms a clustered representation containing groups of neurons that each encode a single task variable (*Figure 5D*, bottom), we may expect different results. Indeed, with a clustered representation, sparse connectivity dramatically reduces performance (*Figure 5E*). This is because the distribution of $\mathbf{Corr}\left(\mathbf{J}_i^{\mathrm{eff}}, \mathbf{J}_j^{\mathrm{eff}}\right)$ deviates substantially from that corresponding to uniform sampling (*Figure 5F*), even as $N \to \infty$ (see Appendix). Specifically, increasing $N$ does not reduce the probability of two expansion layer neurons being connected to input layer neurons that encode the same task variables and therefore receiving highly correlated currents. As a result, expansion layer neurons do not sample task variables uniformly and performance is dramatically reduced.

Our results show that networks with small $K$, moderately large $N$, and a distributed input layer representation approach the performance of networks that sample task variables uniformly. This equivalence validates the applicability of our theory to these more realistic networks. It also argues for the importance of distributed sensorimotor representations in the cortico-cerebellar pathway, consistent with the distributed nature of representations in motor cortex (*Shenoy et al., 2013*; *Muscinelli et al., 2023*).

## Optimal cerebellar architecture is consistent across tasks

A history of theoretical modeling has shown a remarkable correspondence between anatomical properties of the cerebellar cortex and model parameters optimal for learning. These include the in-degree $K$ of granule cells (*Marr, 1969*; *Litwin-Kumar et al., 2017*; *Cayco-Gajic et al., 2017*), the expansion ratio of the granule cells to the mossy fibers $M/N$ (*Babadi and Sompolinsky, 2014*; *Litwin-Kumar et al., 2017*), and the distribution of synaptic weights from granule cells to Purkinje cells (*Brunel et al., 2004*; *Clopath et al., 2012*; *Clopath and Brunel, 2013*). In these studies, model performance was assessed using random categorization tasks. We have shown that optimal coding level is dependent on the task being learned, raising the question of whether optimal values of these architectural parameters are also task-dependent.

Sparse connectivity ($K$=4, consistent with the typical in-degree of cerebellar granule cells) has been shown to optimize learning performance in cerebellar cortex models (*Litwin-Kumar et al., 2017*; *Cayco-Gajic et al., 2017*). We examined the performance of networks with different granule cell in-degrees learning Gaussian process targets. The optimal in-degree is small for all the tasks we consider, suggesting that sparse connectivity is sufficient for high performance across a range of

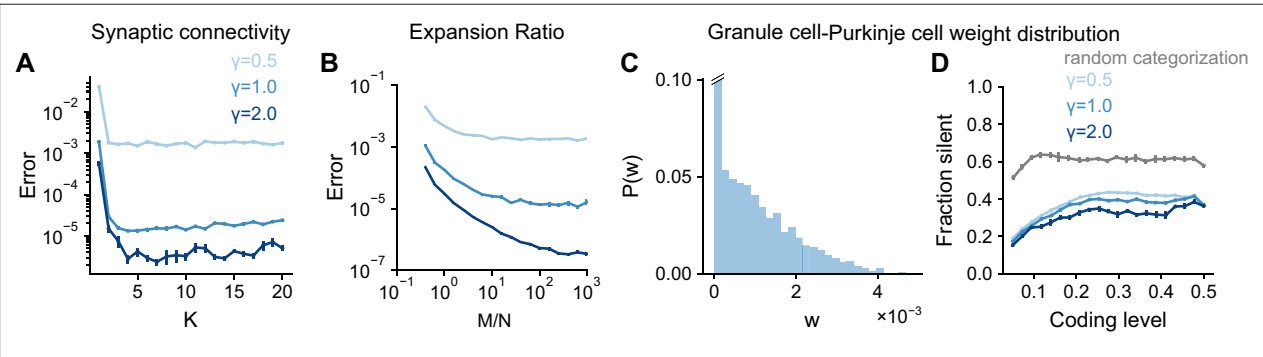

**Figure 6.** Task-independence of optimal anatomical parameters. (**A**) Error as a function of in-degree $K$ for networks learning Gaussian process targets. Curves represent different values of $\gamma$, the length scale of the Gaussian process. The total number of synaptic connections $S = MK$ is held constant. This constraint introduces a trade-off between having many neurons with small synaptic degree and having fewer neurons with large synaptic degree (*Litwin-Kumar et al., 2017*). $S = 10^4$, $D = 3$, $f = 0.3$. (**B**) Error as a function of expansion ratio $M/N$ for networks learning Gaussian process targets. $D = 3$, $N = 700$, $f = 0.3$. (**C**) Distribution of granule-cell-to-Purkinje cell weights $w$ for a network trained on nonnegative Gaussian process targets with $f = 0.3$, $D = 3$, $\gamma = 1$. Granule-cell-to-Purkinje cell weights are constrained to be nonnegative (*Brunel et al., 2004*). (**D**) Fraction of granule-cell-to-Purkinje cell weights that are silent in networks learning nonnegative Gaussian process targets (blue) and random categorization tasks (gray).

tasks (*Figure 6A*). This is consistent with the previous observation that the performance of a sparsely connected network approaches that of a fully connected network (*Figure 5B*).

Previous studies also showed that the expansion ratio from mossy fibers to granule cells $M/N$ controls the dimension of the granule cell representation (*Babadi and Sompolinsky, 2014*; *Litwin-Kumar et al., 2017*). The dimension increases with expansion ratio but saturates as expansion ratio approaches the anatomical value ($M/N \approx 30$ when $f \approx 0.1$ for the inputs presynaptic to an individual Purkinje cell). These studies assumed that mossy fiber activity is uncorrelated ($D = N$) rather than low-dimensional ($D < N$). This raises the question of whether a large expansion is beneficial when $D$ is small. We find that when the number of training patterns $P$ is sufficiently large, performance still improves as $M/N$ approaches its anatomical value, showing that Purkinje cells can exploit their large number of presynaptic inputs even in the case of low-dimensional activity (*Figure 6B*).

*Brunel et al., 2004* showed that the distribution of granule-cell-to-Purkinje cell synaptic weights is consistent with the distribution that maximizes the number of random binary input-output mappings stored. This distribution exhibits a substantial fraction of silent synapses, consistent with experiments. These results also hold for analog inputs and outputs (*Clopath and Brunel, 2013*) and for certain forms of correlations among binary inputs and outputs (*Clopath et al., 2012*). However, the case we consider, where targets are a smoothly varying function of task variables, has not been explored. We observe a similar weight distribution for these tasks (*Figure 6C*), with the fraction of silent synapses remaining high across coding levels (*Figure 6D*). The fraction of silent synapses is lower for networks learning Gaussian process targets than those learning random categorization tasks, consistent with the capacity of a given network for learning such targets being larger (*Clopath et al., 2012*).

Although optimal coding level is task-dependent, these analyses suggest that optimal architectural parameters are largely task-independent. Whereas coding level tunes the inductive bias of the network to favor the learning of specific tasks, these architectural parameters control properties of the representation that improve performance across tasks. In particular, sparse connectivity and a large expansion support uniform sampling of low-dimensional task variables (consistent with *Figure 5C*), while a large fraction of silent synapses is a consequence of a readout that maximizes learning performance (*Brunel et al., 2004*).

## Modeling specific behaviors dependent on cerebellum-like structures

So far, we have considered analytically tractable families of tasks with parameterized input-output functions. Next, we extend our results to more realistic tasks constrained by the functions of specific cerebellum-like systems, which include both highly structured, continuous input-output mappings and random categorization tasks.

To model the cerebellum's role in predicting the consequences of motor commands (*Wolpert et al., 1998*), we examined the optimal coding level for learning the dynamics of a two-joint arm (*Fagg et al., 1997*). Given an initial state, the network predicts the change in the future position of the arm (*Figure 7A*). Performance is optimized at substantially higher coding levels than for random categorization tasks, consistent with our previous results for continuous input-output mappings (*Figure 2E and F*).

The mushroom body, a cerebellum-like structure in insects, is required for learning of associations between odors and appetitive or aversive valence (*Modi et al., 2020*). This behavior can be represented as a mapping from random representations of odors in the input layer to binary category labels (*Figure 7B*). The optimal coding level in a model with parameters consistent with the *Drosophila* mushroom body is less than 0.1, consistent with our previous results for random categorization tasks (*Figure 2E*) and the sparse odor-evoked responses in *Drosophila* Kenyon cells (*Turner et al., 2008*; *Honegger et al., 2011*; *Lin et al., 2014*).

The prediction and cancellation of self-generated sensory feedback has been studied extensively in mormyrid weakly electric fish and depends on the electrosensory lateral line lobe (ELL), a cerebellum-like structure (*Bell et al., 2008*). Granule cells in the ELL provide a temporal basis for generating negative images that are used to cancel self-generated feedback (*Figure 7C*). We extended a detailed model of granule cells and their inputs (*Kennedy et al., 2014*) to study the influence of coding level on the effectiveness of this basis. The performance of this model saturated at relatively high coding levels, and notably the coding level corresponding to biophysical parameters estimated from data

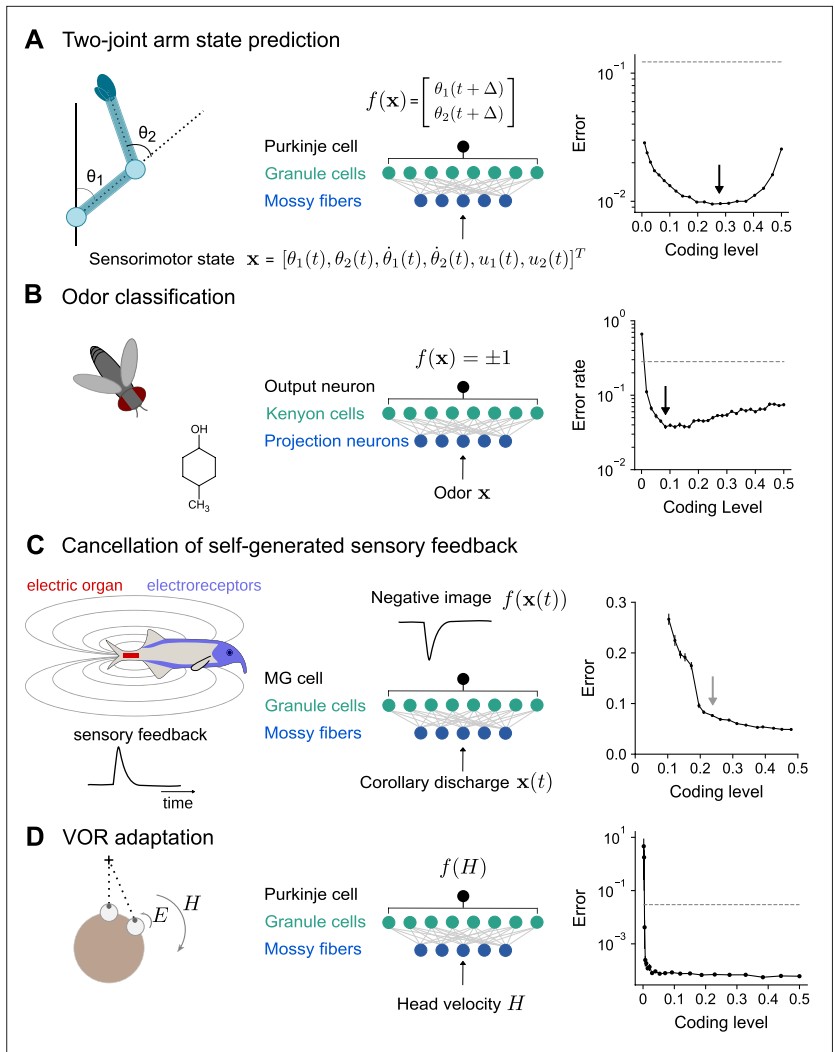

**Figure 7.** Optimal coding level across tasks and neural systems. (**A**) Left: Schematic of two-joint arm. Center: Cerebellar cortex model in which sensorimotor task variables at time $t$ are used to predict hand position at time $t + \delta$. Right: Error as a function of coding level. Black arrow indicates location of optimum. Dashed line indicates performance of a readout of the input layer. (**B**) Left: Odor categorization task. Center: *Drosophila* mushroom body model in which odors activate olfactory projection neurons and are associated with a binary category (appetitive or aversive). Right: Error rate, similar to (**A**), right. (**C**) Left: Schematic of electrosensory system of the mormyrid electric fish, which learns a negative image to cancel the self-generated feedback from electric organ discharges sensed by electroreceptors. Center: Electrosensory lateral line lobe (ELL) model in which MG cells learn a negative image. Right: Error as a function of coding level. Gray arrow indicates location of coding level estimated from biophysical parameters (*Kennedy et al., 2014*). (**D**) Left: Schematic of the vestibulo-cular reflex (VOR). Head rotations with velocity $H$ trigger eye motion in the opposite direction with velocity $E$. During VOR adaptation, organisms adapt to different gains ($E/H$). Center: Cerebellar cortex model in which the target function is the Purkinje cell's firing rate as a function of head velocity. Right: Error, similar to (**A**), right.

The online version of this article includes the following figure supplement(s) for figure 7:

**Figure supplement 1.** Optimal coding levels in the presence of spiking noise.

**Figure supplement 2.** Task-dependence of optimal coding level remains consistent under an online climbing fiber-based plasticity rule.

coincided with the value at which further increases in performance were modest. This observation suggests that coding level is also optimized for task performance in this system.

A canonical function of the mammalian cerebellum is the adjustment of the vestibulo-ocular reflex (VOR), in which motion of the head is detected and triggers compensatory ocular motion in the

opposite direction. During VOR learning, Purkinje cells are tuned to head velocity, and their tuning curves are described as piecewise linear functions (*Lisberger et al., 1994*; *Figure 7D*). Although *in vivo* population recordings of granule cells during VOR adaptation are not, to our knowledge, available for comparison, our model predicts that performance for learning such tuning curves is high across a range of coding levels and shows that sparse codes are sufficient (although not necessary) for such tasks (*Figure 7D*).

These results predict diverse coding levels across different behaviors dependent on cerebellum-like structures. The odor categorization and VOR tasks both have input-output mappings that exhibit sharp nonlinearities and can be efficiently learned using sparse representations. In contrast, the forward modeling and feedback cancellation tasks have smooth input-output mappings and exhibit denser optima. These observations are consistent with our previous finding that more structured tasks favor denser coding levels than do random categorization tasks (*Figure 2E and F*).

## Discussion

We have shown that the optimal granule cell coding level depends on the task being learned. While sparse representations are suitable for learning to categorize inputs into random categories, as predicted by classic theories, tasks involving structured input-output mappings benefit from denser representations (*Figure 2*). This reconciles such theories with the observation of dense granule cell activation during movement (*Knogler et al., 2017*; *Wagner et al., 2017*; *Giovannucci et al., 2017*; *Badura and De Zeeuw, 2017*; *Wagner et al., 2019*). We also show that, in contrast to the task-dependence of optimal coding level, optimal anatomical values of granule cell and Purkinje cell connectivity are largely task-independent (*Figure 6*). This distinction suggests that a stereotyped cerebellar architecture may support diverse representations optimized for a variety of learning tasks.

### Relationship to previous theories

Previous studies assessed the learning performance of cerebellum-like systems with a model Purkinje cell that associates random patterns of mossy fiber activity with one of two randomly assigned categories (*Marr, 1969*; *Albus, 1971*; *Brunel et al., 2004*; *Babadi and Sompolinsky, 2014*; *Litwin-Kumar et al., 2017*; *Cayco-Gajic et al., 2017*), a common benchmark for artificial learning systems (*Gerace et al., 2022*). In this case, a low coding level increases the dimension of the granule cell representation, permitting more associations to be stored and improving generalization to previously unseen inputs. The optimal coding level is low but not arbitrarily low, as extremely sparse representations introduce noise that hinders generalization (*Barak et al., 2013*; *Babadi and Sompolinsky, 2014*).

To examine a broader family of tasks, our learning problems extend previous studies in several ways. First, we consider inputs that may be constrained to a low-dimensional task subspace. Second, we consider input-output mappings beyond random categorization tasks. Finally, we assess generalization error for arbitrary locations on the task subspace, rather than only for noisy instances of previously presented inputs. As we have shown, these considerations require a complete analysis of the inductive bias of cerebellum-like networks (*Figure 4*). Our analysis generalizes previous approaches (*Barak et al., 2013*; *Babadi and Sompolinsky, 2014*; *Litwin-Kumar et al., 2017*) that focused on dimension and noise alone. In particular, both dimension and noise for random patterns can be directly calculated from the kernel function (*Figure 3C*; see Appendix).

Our theory builds upon techniques that been developed for understanding properties of kernel regression (*Sollich, 1998*; *Jacot et al., 2018*; *Bordelon et al., 2020*; *Canatar et al., 2021b*; *Simon et al., 2021*). Kernel approximations of wide neural networks are a major area of current research providing analytically tractable theories (*Rahimi and Recht, 2007*; *Jacot et al., 2018*; *Chizat et al., 2018*). Prior studies have analyzed kernels corresponding to networks with zero (*Cho and Saul, 2010*) or mean-zero Gaussian thresholds (*Basri et al., 2019*; *Jacot et al., 2018*), which in both cases produce networks with a coding level of 0.5. Ours is the first kernel study of the effects of nonzero average thresholds. Our full characterization of the eigenvalue spectra and their decay rates as a function of the threshold extends previous work (*Bach, 2017*; *Bietti and Bach, 2021*). Furthermore, artificial neural network studies typically assume either fully-connected or convolutional layers, yet pruning connections after training barely degrades performance (*Han et al., 2015*; *Zhang et al., 2018*). Our results

support the idea that sparsely connected networks may behave like dense ones if the representation is distributed (*Figure 5*), providing insight into the regimes in which pruning preserves performance.

Other studies have considered tasks with smooth input-output mappings and low-dimensional inputs, finding that heterogeneous Golgi cell inhibition can improve performance by diversifying individual granule cell thresholds (*Spanne and Jörntell, 2013*). Extending our model to include heterogeneous thresholds is an interesting direction for future work. Another proposal states that dense coding may improve generalization (*Spanne and Jörntell, 2015*). Our theory reveals that whether or not dense coding is beneficial depends on the task.

## Assumptions and extensions

We have made several assumptions in our model for the sake of analytical tractability. When comparing the inductive biases of networks with different coding levels, our theory assumes that inputs are normalized and distributed uniformly in a linear subspace of the input layer activity. This allows us to decompose the target function into a basis in which we can directly compare eigenvalues, and hence learning performance, for different coding levels (*Figure 4E–G*). A similar analysis can be performed when inputs are not uniformly distributed, but in this case the basis is determined by an interplay between this distribution and the nonlinearity of expansion layer neurons, making the analysis more complex (see Appendix). We have also assumed that generalization is assessed for inputs drawn from the same distribution as used for learning. Recent and ongoing work on out-of-distribution generalization may permit relaxations of this assumption (*Shen et al., 2021*; *Canatar et al., 2021a*).

When analyzing properties of the granule cell layer, our theory also assumes an infinitely wide expansion. When $P$ is small enough that performance is limited by number of samples, this assumption is appropriate, but finite-size corrections to our theory are an interesting direction for future work. We also have not explicitly modeled inhibitory input provided by Golgi cells, instead assuming such input can be modeled as a change in effective threshold, as in previous studies (*Billings et al., 2014*; *Cayco-Gajic et al., 2017*; *Litwin-Kumar et al., 2017*). This is appropriate when considering the dimension of the granule cell representation (*Litwin-Kumar et al., 2017*), but more work is needed to extend our model to the case of heterogeneous inhibition.

Another key assumption concerning the granule cells is that they sample mossy fiber inputs randomly, as is typically assumed in Marr-Albus models (*Marr, 1969*; *Albus, 1971*; *Litwin-Kumar et al., 2017*; *Cayco-Gajic et al., 2017*). Other studies instead argue that granule cells sample from mossy fibers with highly similar receptive fields (*Garwicz et al., 1998*; *Brown and Bower, 2001*; *Jörntell and Ekerot, 2006*) defined by the tuning of mossy fiber and climbing fiber inputs to cerebellar microzones (*Apps et al., 2018*). This has led to an alternative hypothesis that granule cells serve to relay similarly tuned mossy fiber inputs and enhance their signal-to-noise ratio (*Jörntell and Ekerot, 2006*; *Gilbert and Chris Miall, 2022*) rather than to re-encode inputs. Another hypothesis is that granule cells enable Purkinje cells to learn piece-wise linear approximations of nonlinear functions (*Spanne and Jörntell, 2013*). However, several recent studies support the existence of heterogeneous connectivity and selectivity of granule cells to multiple distinct inputs at the local scale (*Huang et al., 2013*; *Ishikawa et al., 2015*). Furthermore, the deviation of the predicted dimension in models constrained by electron-microscopy data as compared to randomly wired models is modest (*Nguyen et al., 2023*). Thus, topographically organized connectivity at the macroscopic scale may coexist with disordered connectivity at the local scale, allowing granule cells presynaptic to an individual Purkinje cell to sample heterogeneous combinations of the subset of sensorimotor signals relevant to the tasks that Purkinje cell participates in. Finally, we note that the optimality of dense codes for learning slowly varying tasks in our theory suggests that observations of a lack of mixing (*Jörntell and Ekerot, 2002*) for such tasks are compatible with Marr-Albus models, as in this case nonlinear mixing is not required.

We have quantified coding level by the fraction of neurons that are above firing threshold. We focused on coding levels $f < 0.5$, as extremely dense codes are rarely found in experiments (*Olshausen and Field, 2004*), but our theory applies for $f > 0.5$ as well. In general, representations with coding levels of $f$ and $1 - f$ perform similarly in our model due to the symmetry of most of their associated eigenvalues (*Figure 4—figure supplement 1* and Appendix). Under the assumption that the energetic costs associated with neural activity are minimized, the $f < 0.5$ region is likely the biologically plausible one. We also note that coding level is most easily defined when neurons are modeled as rate, rather than spiking units. To investigate the consistency of our results under a spiking code, we

implemented a model in which granule cell spiking exhibits Poisson variability and quantify coding level as the fraction of neurons that have nonzero spike counts (*Figure 7—figure supplement 1*; *Figure 7C*). In general, increased spike count leads to improved performance as noise associated with spiking variability is reduced. Granule cells have been shown to exhibit reliable burst responses to mossy fiber stimulation (*Chadderton et al., 2004*), motivating models using deterministic responses or sub-Poisson spiking variability. However, further work is needed to quantitatively compare variability in model and experiment and to account for more complex biophysical properties of granule cells (*Saarinen et al., 2008*).

For the Purkinje cells, our model assumes that their responses to granule cell input can be modeled as an optimal linear readout. Our model therefore provides an upper bound to linear readout performance, a standard benchmark for the quality of a neural representation that does not require assumptions on the nature of climbing fiber-mediated plasticity, which is still debated. Electrophysiological studies have argued in favor of a linear approximation (*Brunel et al., 2004*). To improve the biological applicability of our model, we implemented an online climbing fiber-mediated learning rule and found that optimal coding levels are still task-dependent (*Figure 7—figure supplement 2*). We also note that although we model several timing-dependent tasks (*Figure 7*), our learning rule does not exploit temporal information, and we assume that temporal dynamics of granule cell responses are largely inherited from mossy fibers. Integrating temporal information into our model is an interesting direction for future investigation.

## Implications for cerebellar representations

Our results predict that qualitative differences in the coding levels of cerebellum-like systems, across brain regions or across species, reflect an optimization to distinct tasks (*Figure 7*). However, it is also possible that differences in coding level arise from other physiological differences between systems. In the *Drosophila* mushroom body, which is required for associative learning of odor categories, random and sparse subsets of Kenyon cells are activated in response to odor stimulation, consistent with our model (*Figure 7B*; *Turner et al., 2008*; *Honegger et al., 2011*; *Lin et al., 2014*). In a model of the electrosensory system of the electric fish, the inferred coding level of a model constrained by the properties of granule cells is similar to that which optimizes task performance (*Figure 7C*). Within the cerebellar cortex, heterogeneity in granule cell firing has been observed across cerebellar lobules, associated with both differences in intrinsic properties (*Heath et al., 2014*) and mossy fiber input (*Witter and De Zeeuw, 2015*). It would be interesting to correlate such physiological heterogeneity with heterogeneity in function across the cerebellum. Our model predicts that regions involved in behaviors with substantial low-dimensional structure, for example smooth motor control tasks, may exhibit higher coding levels than regions involved in categorization or discrimination of high-dimensional stimuli.

Our model also raises the possibility that individual brain regions may exhibit different coding levels at different moments in time, depending on immediate behavioral or task demands. Multiple mechanisms could support the dynamic adjustment of coding level, including changes in mossy fiber input (*Ozden et al., 2012*), Golgi cell inhibition (*Eccles et al., 1966*; *Palay and Chan-Palay, 1974*), retrograde signaling from Purkinje cells (*Kreitzer and Regehr, 2001*), or unsupervised plasticity of mossy fiber-to-granule cell synapses (*Schweighofer et al., 2001*). The predictions of our model are not dependent on which of these mechanisms are active. A recent study demonstrated that local synaptic inhibition by Golgi cells controls the spiking threshold and hence the population coding level of cerebellar granule cells in mice (*Fleming et al., 2022*). Further, the authors observed that granule cell responses to sensory stimuli are sparse when movement-related selectivity is controlled for. This suggests that dense movement-related activity and sparse sensory-evoked activity are not incompatible.

While our analysis makes clear qualitative predictions concerning comparisons between the optimal coding levels for different tasks, in some cases it is also possible to make quantitative predictions about the location of the optimum for a single task. Doing so requires determining the appropriate time interval over which to measure coding level, which depends on the integration time constant of the readout neuron. It also requires estimates of the firing rates and biophysical properties of the expansion layer neurons. In the electrosensory system, for which a well-calibrated model exists and the learning objective is well-characterized (*Kennedy et al., 2014*), we found

that the coding level estimated based on the data is similar to that which optimizes performance (*Figure 7C*).

If coding level is task-optimized, our model predicts that manipulating coding level artificially will diminish performance. In the *Drosophila* mushroom body, disrupting feedback inhibition from the GABAergic anterior paired lateral neuron onto Kenyon cells increases coding level and impairs odor discrimination (*Lin et al., 2014*). A recent study demonstrated that blocking inhibition from Golgi cells onto granule cells results in denser granule cell population activity and impairs performance on an eye-blink conditioning task (*Fleming et al., 2022*). These examples demonstrate that increasing coding level during sensory discrimination tasks, for which sparse activity is optimal, impairs performance. Our theory predicts that decreasing coding level during a task for which dense activity is optimal, such as smooth motor control, would also impair performance.

While dense activity has been taken as evidence against theories of combinatorial coding in cerebellar granule cells (*Knogler et al., 2017*; *Wagner et al., 2019*), our theory suggests that the two are not incompatible. Instead, the coding level of cerebellum-like regions may be determined by behavioral demands and the nature of the input to granule-like layers (*Muscinelli et al., 2023*). Sparse coding has also been cited as a key property of sensory representations in the cerebral cortex (*Olshausen and Field, 1996*). However, recent population recordings show that such regions exhibit dense movement-related activity (*Musall et al., 2019*), much like in cerebellum. While the theory presented in this study does not account for the highly structured recurrent interactions that characterize cerebrocortical regions, it is possible that these areas also operate using inductive biases that are shaped by coding level in a similar manner to our model.

## Methods
### Network model
The expansion layer activity is given by $\mathbf{h} = \phi(\mathbf{J}^{\text{eff}}\mathbf{x} - \theta)$, where $\mathbf{J}^{\text{eff}} = \mathbf{J}\mathbf{A}$ describes the selectivity of expansion layer neurons to task variables. For most simulations, $\mathbf{A}$ is an $N \times D$ matrix sampled with random, orthonormal columns and $\mathbf{J}$ is an $M \times N$ matrix with i.i.d. unit Gaussian entries. The nonlinearity $\phi$ is a rectified linear activation function $\phi(u) = \max(u, 0)$ applied element-wise. The input layer activity $\mathbf{n}$ is given by $\mathbf{n} = \mathbf{A}\mathbf{x}$.

### Sparsely connected networks
To model sparse excitatory connectivity, we generated a sparse matrix $\mathbf{J}^E$, where each row contains precisely $K$ nonzero elements at random locations. The nonzero elements are either identical and equal to 1 (homogeneous excitatory weights) or sampled from a unit truncated normal distribution (heterogeneous excitatory weights). To model global feedforward inhibition that balances excitation, $\mathbf{J} = \mathbf{J}^E - \mathbf{J}^I$, where $\mathbf{J}^I$ is a dense matrix with every element equal to $\frac{1}{MN}\sum_{ij}J^E_{ij}$.

For *Figure 5B*, *Figure 6A and B*, *Figure 7B*, sparsely connected networks were generated with homogeneous excitatory weights and global inhibition. For *Figure 5E*, the network with clustered representations was generated with homogeneous excitatory weights without global inhibition. For *Figure 5C and F*, networks were generated with heterogeneous excitatory weights and global inhibition.

### Clustered representations
For clustered input-layer representations, each input layer neuron encodes one task variable (that is, $\mathbf{A}$ is a block matrix, with nonoverlapping blocks of $N/D$ elements equal to 1 for each task variable). In this case, in order to obtain good performance, we found it necessary to fix the coding level for each input pattern, corresponding to winner-take-all inhibition across the expansion layer.

### Dimension
The dimension of the expansion layer representation (*Figure 3D*) is given by *Abbott et al., 2011*; *Litwin-Kumar et al., 2017*:

$$d = \frac{(\sum_i \lambda_i)^2}{(\sum_i \lambda_i^2)}, \tag{5}$$

where $\lambda_i$ are the eigenvalues of the covariance matrix $C_{ij}^{\mathbf{h}} = \mathrm{Cov}(h_i, h_j)$ of expansion layer responses (not to be confused with $\lambda_\alpha$, the eigenvalues of the kernel operator). The covariance is computed by averaging over inputs $\mathbf{x}$.

## Learning tasks

### Random categorization task

In a random categorization task (*Figure 2E*, *Figure 7B*), the network learns to associate a random input pattern $\mathbf{x}^\mu \in \mathbb{R}^D$ for $\mu = 1, \ldots, P$ with a random binary category $y^\mu = \pm 1$. The elements of $\mathbf{x}^\mu$ are drawn i.i.d. from a normal distribution with mean 0 and variance $1/D$. Test patterns $\hat{\mathbf{x}}^\mu$ are generated by adding noise to the training patterns:

$$\hat{\mathbf{x}}^\mu = \sqrt{1 - \epsilon^2}\mathbf{x}^\mu + \epsilon\boldsymbol{\eta}, \tag{6}$$

where $\boldsymbol{\eta} \sim \mathcal{N}(0, \frac{1}{D}\mathbf{I})$. For *Figure 2E*, *Figure 7B*, and *Figure 4—figure supplement 2*, we set $\epsilon = 0.1$.

### Gaussian process tasks

To generate a family of tasks with continuously varying outputs (*Figure 2D and F*, *Figure 4F and G*, *Figure 5B*, and *Figure 6*), we sampled target functions from a Gaussian process (*Rasmussen and Williams, 2006*), $f(\mathbf{x}) \sim \mathcal{GP}(0, C)$, with covariance

$$C(\mathbf{x}^\mu, \mathbf{x}^\nu) = \exp\left(-\frac{1}{2\gamma^2}\|\mathbf{x}^\mu - \mathbf{x}^\nu\|^2\right), \tag{7}$$

where $\gamma$ determines the spatial scale of variations in $f(\mathbf{x})$. Training and test patterns are drawn uniformly on the unit sphere.

### Learning of readout weights

With the exception of the ELL task and *Figure 7—figure supplement 2*, we performed unregularized least squares regression to determine the readout weights $\mathbf{w}$. For the ELL sensory cancellation task (*Figure 7C*), we used $\ell^2$ regularization, a.k.a. ridge regression:

$$\mathbf{w} = \underset{\mathbf{w}'}{\mathbf{argmin}} \sum_{\mu=1}^{P} \|f(\mathbf{x}^\mu) - \mathbf{w}' \cdot \mathbf{h}(\mathbf{x}^\mu)\|^2 + M\alpha_{\mathbf{ridge}}\|\mathbf{w}'\|_2^2, \tag{8}$$

where $\alpha_{\mathrm{ridge}}$ is the regularization parameter. Solutions were found using Python's scikit-learn package (*Pedregosa, 2011*).

In *Figure 7—figure supplement 2*, we implement a model of an online climbing fiber-mediated plasticity rule. The climbing fiber activity $c$ is assumed to encode the error between the target and the network prediction $c = f(\mathbf{x}) - \hat{f}(\mathbf{x})$. During each of $N_{\mathrm{epochs}}$ training epochs, the $P$ training patterns are shuffled randomly and each pattern is presented one at a time. For each pattern µ, the weights are updated according to $\Delta\mathbf{w}^\mu = \eta \cdot c \cdot \mathbf{h}(\mathbf{x}^\mu)$. Parameter values were $P = 30, \eta = 0.7/M, M = 10{,}000, N_{\mathrm{epochs}} = 20{,}000$.

### Performance metrics

For tasks with continuous targets, the prediction of the network is given by $\hat{f}(\mathbf{x}) = \mathbf{w} \cdot \mathbf{h}(\mathbf{x})$, where $\mathbf{w}$ are the synaptic weights of the readout from the expansion layer. Error is measured as relative mean squared error (an expectation across patterns $\mathbf{x}$ in the test set): $\mathrm{Error} = \frac{\mathbb{E}[(f(\mathbf{x}) - \hat{f}(\mathbf{x}))^2]}{\mathbf{x}[f(\mathbf{x})^2]}$. In practice we use a large test set to estimate this error over $\mathbf{x}$ drawn from the distribution of test patterns. For categorization tasks, the network's prediction is given by $\hat{f}(\mathbf{x}) = \mathrm{sign}(\mathbf{w} \cdot \mathbf{h}(\mathbf{x}))$. Performance is measured as the fraction of incorrect predictions. Error bars represent standard error of the mean across realizations of network weights and tasks.

## Optimal granule–Purkinje cell weight distribution

We adapted our model to allow for comparisons with *Brunel et al., 2004* by constraining readout weights **w** to be nonnegative and adding a bias, $f(\mathbf{x}) = \mathbf{w} \cdot \mathbf{h}(\mathbf{x}) + b$. To guarantee that the target function is nonnegative, we set $f(\mathbf{x}) \in \{0, 1\}$ for the random categorization task and $f(\mathbf{x}) \leftarrow |f(\mathbf{x})|$ for the Gaussian process tasks. The weights and bias were determined with the Python convex optimization package cvxopt (*Andersen et al., 2011*).

## Model of two-joint arm

We implemented a biophysical model of a planar two-joint arm (*Fagg et al., 1997*). The state of the arm is specified by six variables: joint angles $\theta_1$ and $\theta_2$, angular velocities $\dot{\theta}_1$ and $\dot{\theta}_2$, and torques $u_1$ and $u_2$. The upper and lower segments of the arm have lengths $l_1$ and $l_2$ and masses $m_1$ and $m_2$, respectively. The arm has the following dynamics:

$$M(\boldsymbol{\theta})\ddot{\boldsymbol{\theta}} + C(\boldsymbol{\theta}, \dot{\boldsymbol{\theta}})\dot{\boldsymbol{\theta}} = \mathbf{u}, \tag{9}$$

where $\mathbf{M}(\boldsymbol{\theta})$ is the inertia matrix and $\mathbf{C}(\boldsymbol{\theta}, \dot{\boldsymbol{\theta}})$ is the matrix of centrifugal, Coriolis, and friction forces:

$$M(\boldsymbol{\theta}) = \begin{pmatrix} I_1 + I_2 + m_2 l_1^2 + 2m_2 l_1 \bar{l}_2 \cos(\theta_2) & I_2 + m_2 l_1 \bar{l}_2 \cos(\theta_2) \\ I_2 + m_2 l_1 l_2 \cos(\theta_2) & I_2 \end{pmatrix}, \tag{10}$$

$$C(\boldsymbol{\theta}, \dot{\boldsymbol{\theta}}) = m_2 l_1 l_2 \sin(\theta_2) \begin{pmatrix} -2\dot{\theta}_2 & -\dot{\theta}_2 \\ \dot{\theta}_1 & 0 \end{pmatrix} + \begin{pmatrix} D_1 & 0 \\ 0 & D_2 \end{pmatrix}, \tag{11}$$

where $\bar{l}_2$ is the center of mass of the lower arm, $I_1$ and $I_2$ are moments of inertia and $D_1$ and $D_2$ are friction terms of the upper and lower arm respectively. These parameters were $m_1 = 3$ kg, $m_2 = 2.5$ kg, $l_1 = 0.3$ m, $l_2 = 0.35$ m, $\bar{l}_2 = 0.21$ m, $I_1 = 0.1$ kg m$^2$, $I_2 = 0.12$ kg m$^2$, $D_1 = 0.05$ kg m$^2$/s and $D_2 = 0.01$ kg m$^2$/s.

The task is to predict the position of the hand based on the forward dynamics of the two-joint arm system, given the arm initial condition and the applied torques. More precisely, the $P$ network inputs $\mathbf{x}^\mu$ were generated by sampling 6-dimensional Gaussian vectors with covariance matrix $\mathbf{C} = \mathrm{diag}(\sigma_\theta, \sigma_\theta, \sigma_{\dot{\theta}}, \sigma_{\dot{\theta}}, \sigma_u, \sigma_u)$, to account for the fact that angles, angular velocities and torques might vary on different scales across simulations. For our results, we used $\sigma_\theta = \sigma_{\dot{\theta}} = 0.1$ and $\sigma_u = 1$. Each sample $\mathbf{x}^\mu$ was then normalized and used to generate initial conditions of the arm, by setting $\theta_1^\mu = \frac{\pi}{4} + x_1^\mu$, $\theta_2^\mu = \frac{\pi}{4} + x_2^\mu$, $\dot{\theta}_1^\mu = x_3^\mu$, and $\dot{\theta}_2^\mu = x_4^\mu$. Torques were generated by setting $u_1^\mu = x_5^\mu$ and $u_2^\mu = x_6^\mu$. The target was constructed by running the dynamics of the arm forward in time for a time $\delta = 0.2$ s, and by computing the difference in position of the "hand" (i.e. the end of the lower segment) in Cartesian coordinates. As a result, the target in this task is two-dimensional, with each target dimension corresponding the one of the two Cartesian coordinates of the hand. The overall performance is assessed by computing the error on each task separately and then averaging the errors.

## Model of electrosensory lateral line lobe (ELL)

We simulated 20,000 granule cells using the biophysical model of *Kennedy et al., 2014*. We varied the granule cell layer coding level by adjusting the spiking threshold parameter in the model. For each choice of threshold, we generated 30 different trials of spike rasters. Each trial is 160ms long with a 1ms time bin and consists of a time-locked response to an electric organ discharge command. Trial-to-trial variability in the model granule cell responses arises from noise in the mossy fiber responses. To generate training and testing data, we sampled 4 trials ($P = 640$ patterns) from the 30 total trials for training and 10 trials for testing (1600 patterns). Coding level is measured as the fraction of granule cells that spike at least once in the training data. We repeated this sampling process 30 times.

The targets were smoothed broad-spike responses of 15 MG cells time-locked to an electric organ discharge command measured during experiments (*Muller et al., 2019*). The original data set consisted of 55 MG cells, each with a 300ms long spike raster with a 1ms time bin. The spike rasters were trial-averaged and then smoothed with a Gaussian-weighted moving average with a 10ms time window. Only MG cells whose maximum spiking probability across all time bins exceeded 0.01 after smoothing were included in the task. The same MG cell responses were used for both training and

testing. To match the length of the granule cell data, we discarded MG cell data beyond 160ms and then concatenated 4 copies of the 160ms long responses for training and 10 copies for testing. We measured the ability of the model to construct MG cell targets out of granule cell activity, generalizing across noise in granule cell responses. Errors for each MG cell target were averaged across the 30 repetitions of sampling of training and testing data, and then averaged across targets. Standard error of the mean was computed across the 30 repetitions.

## Model of vestibulo-ocular reflex (VOR)

Recordings of Purkinje cell activity in monkeys suggest that these neurons exhibit piecewise-linear tuning to head velocity (*Lisberger et al., 1994*). Thus, we designed piecewise-linear target functions representing Purkinje cell firing rate as a function of head velocity $v$, a one-dimensional input:

$$f(v) = \begin{cases} m_1(v-c) + b & x < c \\ m_2(v-c) + b & x \geq c. \end{cases} \tag{12}$$

Inputs $v$ were sampled uniformly from $[-1, 1]$ 100 times. We generated 25 total target functions using all combinations of slopes $m_1$ and $m_2$ sampled from 5 equally spaced points on the interval $[-2, 2]$. We set $b = 0.1$ and $c = -0.2$.

Mossy fiber responses to head velocity input were modeled as exponential tuning curves:

$$n_j(v) = g_j \exp(vr_j) + b_j, \tag{13}$$

where $g_j$ is a gain term, $r_j \in \pm 1$ determines a mossy fiber preference for positive or negative velocities, and $b_j$ is the baseline firing rate. We generated 24 different tuning curves from all combinations of the following parameter values: The gain $g_j$ was sampled from 6 equally spaced points on the interval $[0.1, 1]$, $r_j$ was set to either –1 or 1, and $b_j$ was set to either 0 or 1. Qualitative results did not depend strongly on this parameterization. Mossy fiber to granule cell weights were random zero-mean Gaussians. Errors were averaged across targets.

## Acknowledgements

The authors thank L F Abbott, N A Cayco-Gajic, N Sawtell, and S Fusi for helpful discussions and comments on the manuscript. The authors also thank S Muller for contributing data, code, and helpful discussions for analysis of ELL data. M X was supported by NIH grant T32-NS064929. S M was supported by the Simons and Swartz Foundations. K D H was supported by a grant from the Washington Research Foundation. A L-K was supported by the Simons and Burroughs Wellcome Foundations. M X, S M, and A L-K were also supported by the Gatsby Charitable Foundation and NSF NeuroNex award DBI-1707398.

## Additional information

### Funding

| Funder | Grant reference number | Author |
|---|---|---|
| National Institutes of Health | T32-NS06492 | Marjorie Xie |
| Simons Foundation | | Samuel P Muscinelli |
| Swartz Foundation | | Samuel P Muscinelli |
| Washington Research Foundation | | Kameron Decker Harris |
| Burroughs Wellcome Fund | | Ashok Litwin-Kumar |
| Gatsby Charitable Foundation | GAT3708 | Marjorie Xie |

| Funder | Grant reference number | Author |
| --- | --- | --- |
| National Science Foundation | DBI-1707398 | Marjorie Xie |

The funders had no role in study design, data collection and interpretation, or the decision to submit the work for publication.

## Author contributions

Marjorie Xie, Conceptualization, Investigation, Writing - original draft, Writing – review and editing; Samuel P Muscinelli, Kameron Decker Harris, Ashok Litwin-Kumar, Conceptualization, Investigation, Writing – review and editing

## Author ORCIDs

Marjorie Xie http://orcid.org/0000-0003-1456-4811
Samuel P Muscinelli https://orcid.org/0000-0002-5256-2289
Kameron Decker Harris http://orcid.org/0000-0002-3716-6173
Ashok Litwin-Kumar http://orcid.org/0000-0003-2422-6576

## Decision letter and Author response

Decision letter https://doi.org/10.7554/eLife.82914.sa1
Author response https://doi.org/10.7554/eLife.82914.sa2

# Additional files

## Supplementary files

• MDAR checklist

## Data availability

The current manuscript is a computational study, so no data have been generated for this manuscript. Code implementing the model is available on github: https://github.com/marjoriexie/cerebellar-task-dependent (copy archived at *Xie, 2023*).

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

# Appendix 1

## 1 Connection between kernel and previous theories

Previous theories (**Babadi and Sompolinsky, 2014**; **Litwin-Kumar et al., 2017**) studied generalization performance for random clusters of inputs associated with binary targets, where test patterns are formed by adding noise to training patterns (**Figure 4A**). The readout is trained using a supervised Hebbian rule with mean-subtracted expansion layer responses, $\mathbf{w} = \sum_\mu y^\mu (\mathbf{h}^\mu - \bar{\mathbf{h}})$, with $\bar{\mathbf{h}} = \frac{1}{P}\sum_{\mu=1}^{P} \mathbf{h}^\mu$. The net input to a readout in response to a test pattern $\hat{\mathbf{h}}^\mu$ from cluster μ is $g^\mu = \mathbf{w} \cdot (\hat{\mathbf{h}}^\mu - \bar{\mathbf{h}})$. The statistics of $g^\mu$ determine generalization performance. For a Hebbian readout, the error rate is expressed in terms of the signal-to-noise ratio (SNR) (**Babadi and Sompolinsky, 2014**):

$$P(\text{Error}) = \frac{1}{2}\text{erfc}\left(\sqrt{\text{SNR}/2}\right). \tag{A1}$$

SNR is given in terms of the mean and variance of $g^\mu$:

$$\text{SNR} = \frac{\left(\mathbb{E}_\mu[y^\mu g^\mu]\right)^2}{\text{Var}(g^\mu)}. \tag{A2}$$

The numerator of SNR is proportional to the average overlap of the expansion layer representations of training and test patterns belonging to the same cluster, which can be expressed in terms of the kernel function $K$:

$$\mathbb{E}_\mu\left[y^\mu g^\mu\right] = \mathbb{E}_\mu\left[(\hat{\mathbf{h}}^\mu - \bar{\mathbf{h}}) \cdot (\mathbf{h}^\mu - \bar{\mathbf{h}})\right] = M\mathbb{E}_\mu\left[K(\hat{\mathbf{x}}^\mu, \mathbf{x}^\mu)\right] - \bar{\mathbf{h}} \cdot \bar{\mathbf{h}}. \tag{A3}$$

For large networks with Gaussian i.i.d. expansion weights, $K(\hat{\mathbf{x}}^\mu, \mathbf{x}^\mu) = K(t)$, where $t = \hat{\mathbf{x}}^\mu \cdot \mathbf{x}^\mu$, and the above equation reduces to $MK(t_{\text{train/test}}) - \bar{\mathbf{h}} \cdot \bar{\mathbf{h}}$, where $t_{\text{train/test}}$ is the typical overlap of training and test patterns belonging to the same cluster. When $\mathbf{x}^\mu \cdot \mathbf{x}^\mu = 1$, $t_{\text{train/test}}$ can be written as $t_{\text{train/test}} = 1 - \Delta$, where $\Delta$ is a measure of within-cluster noise (**Babadi and Sompolinsky, 2014**; **Litwin-Kumar et al., 2017**).

**Babadi and Sompolinsky, 2014** demonstrated that, for random categorization tasks and when $M$ and $D$ are large, $\text{Var}\left(g^\mu\right) = C(\frac{1}{M} + Q^2 \frac{1}{D})$ where $C$ is a constant and $Q \in [0, 1]$ is given by

$$Q^2 = \frac{\frac{1}{Z_h}\mathbb{E}_{\mu \neq \nu}\left[((\mathbf{h}^\mu - \bar{\mathbf{h}}) \cdot (\mathbf{h}^\nu - \bar{\mathbf{h}}))^2\right]}{\frac{1}{Z_x}\mathbb{E}_{\mu \neq \nu}\left[(\mathbf{x}^\mu \cdot \mathbf{x}^\nu)^2\right]}, \tag{A4}$$

assuming the entries of $\mathbf{x}$ are zero-mean. $Z_a = \mathbb{E}_\mu[\|\mathbf{a}^\mu - \bar{\mathbf{a}}\|^2]$ for $a \in \{h, x\}$ normalizes the overlaps to the typical overlap of a pattern with itself. The quantity $Q^2$ is the ratio of the variance of overlaps between patterns belonging to different clusters in the expansion layer to that of the input layer. This describes the extent to which the geometry of the input layer representation is preserved in the expansion layer. When overlaps in the input layer are small, as they are for random clusters, $\frac{1}{Z_h}(\mathbf{h}^\mu - \bar{\mathbf{h}}) \cdot (\mathbf{h}^\nu - \bar{\mathbf{h}}) \approx \frac{Q}{Z_x} \cdot (\mathbf{x}^\mu \cdot \mathbf{x}^\nu)$ as $M \to \infty$. This relation illustrates that, for random clusters and $M \to \infty$, $Q$ is equal to the slope of the normalized kernel function $K(t)$ evaluated at $t = 0$. **Litwin-Kumar et al., 2017** also showed that the dimension of the expansion layer representation is equal to $\frac{C'}{(\frac{1}{M} + Q^2 \frac{1}{D})}$, where $C'$ is a constant.

Thus, for the random categorization task studied in **Babadi and Sompolinsky, 2014**; **Litwin-Kumar et al., 2017**, dimension and readout SNR can be calculated by evaluating $K(t_{\text{train/test}})$ and the slope of $K(t)$ at $t = 0$.

## 2 Dot-product kernels with arbitrary threshold

As $M \to \infty$, the normalized dot product between features (**Equation 2**) converges pointwise to

$$K(\mathbf{x}, \mathbf{x}') = \mathbb{E}_{\mathbf{J}_1}\left[\phi(\mathbf{J}_1^T \mathbf{x} - \theta)\phi(\mathbf{J}_1^T \mathbf{x}' - \theta)\right], \tag{A5}$$

where $\mathbf{J}_1$ is a row of the weight matrix $\mathbf{J}$ (without loss of generality, the first row) with entries drawn i.i.d. from a Gaussian distribution $\mathcal{N}(0, 1)$. Our goal is to compute *Equation A5* for a given $\theta$ and inputs drawn on the unit sphere $\mathbf{x}, \mathbf{x}' \in \mathbb{S}^{D-1}$.

Because the Gaussian weight distribution is spherically symmetric, *Equation A5* restricted to the unit sphere *for any nonlinearity* is only a function of the dot-product $t := \mathbf{x}^T \mathbf{x}'$, making the kernel a dot-product kernel $K(\mathbf{x}, \mathbf{x}') = K(t)$.

Denote by $J_i$ the entries of $\mathbf{J}_1$. Let $I_1 = \sum_{i=1}^{D} J_i x_i$ and $I_2 = \sum_{i=1}^{D} J_i x_i'$ be the pre-activations for each input. Then $(I_1, I_2)$ are jointly Gaussian with mean 0, variance 1, and covariance $\mathbb{E}[I_1 I_2] = t$. If $t > 0$, we can re-parameterize these pre-activations as the sum of an independent and shared component $I_i = y_i \sqrt{1-t} + z\sqrt{t}$, where $y_i \sim \mathcal{N}(0, 1)$ for $i = 1, 2$ and $z \sim \mathcal{N}(0, 1)$. In these coordinates, *Equation A5* becomes

$$
\begin{aligned}
K(t) &= \mathop{\mathbb{E}}_{y_1, y_2, z} \left[ \phi(y_1\sqrt{1-t} + z\sqrt{t} - \theta)\phi(y_2\sqrt{1-t} + z\sqrt{t} - \theta) \right] \\
&= \mathop{\mathbb{E}}_{z} \left[ \mathop{\mathbb{E}}_{y_1}[\phi(y_1\sqrt{1-t} + z\sqrt{t} - \theta)|z] \mathop{\mathbb{E}}_{y_2}[\phi(y_2\sqrt{1-t} + z\sqrt{t} - \theta)|z] \right] \\
&= \mathop{\mathbb{E}}_{z} \left[ \mathop{\mathbb{E}}_{y_1}[\phi(y_1\sqrt{1-t} + z\sqrt{t} - \theta)|z]^2 \right],
\end{aligned}
\tag{A6}
$$

where the second line follows from the conditional independence of $h_1|z$ and $h_2|z$ and the third from the fact that they are identically distributed. Similarly, if $t < 0$, we can write $I_1 = y_1\sqrt{1-t} + z\sqrt{t}$, $I_2 = y_2\sqrt{1-t} - z\sqrt{t}$.

We will use *Equation A6* to solve for the kernel assuming $\phi$ is a ReLU nonlinearity. Let

$$
g_1(t, z) = \mathbb{E}[\phi(y_1\sqrt{1-t} + z\sqrt{t} - \theta)|z].
\tag{A7}
$$

Using the fact that $\phi$ is nonzero only when $y_1\sqrt{1-t} + z\sqrt{t} - \theta > 0$, i.e. for $y_1 > T = \frac{\theta - z\sqrt{t}}{\sqrt{1-t}}$, we obtain

$$
\begin{aligned}
g_1(t, z) &= (2\pi)^{-1/2} \int_T^\infty (y_1\sqrt{1-t} + z\sqrt{t} - \theta)e^{-y_1^2/2} \mathrm{d}y_1 \\
&= \left(\frac{1-t}{2\pi}\right)^{1/2} e^{-T^2/2} + \left(\frac{z\sqrt{t} - \theta}{2}\right) \mathrm{erfc}(T/\sqrt{2}).
\end{aligned}
\tag{A8}
$$

Performing a similar calculation for $t < 0$ and collecting the results leads to:

$$
K(t) = \begin{cases} \mathbb{E}_z\left[g_1(t, z)^2\right] & t > 0 \\ \mathbb{E}_z\left[g_1(|t|, z)g_2(|t|, z)\right] & t < 0 \end{cases},
\tag{A9}
$$

$$
g_1(t, z) = \left(\frac{1-t}{2\pi}\right)^{1/2} e^{-T^2/2} + \left(\frac{z\sqrt{t} - \theta}{2}\right) \mathrm{erfc}(T_1/\sqrt{2})
\tag{A10}
$$

$$
g_2(t, z) = \left(\frac{1-t}{2\pi}\right)^{1/2} e^{-T^2/2} + \left(\frac{-z\sqrt{t} - \theta}{2}\right) \mathrm{erfc}(T_2/\sqrt{2})
\tag{A11}
$$

$$
T_1 = \frac{\theta - z\sqrt{t}}{\sqrt{1-t}}, \quad \text{and} \quad T_2 = \frac{\theta + z\sqrt{t}}{\sqrt{1-t}}.
\tag{A12}
$$

## 3 Spherical harmonic decompositions

Our theory of generalization requires us to work in function spaces which are natural to the problem. The spherical harmonics are the natural basis for working with dot-product kernels on the sphere. For a thorough treatment of spherical harmonics, see *Atkinson and Han, 2012*, whose notation we generally follow. Both our kernel and Gaussian process (GP) tasks are defined over the sphere in $D$ dimensions

$$
\mathbb{S}^{D-1} = \{\mathbf{x} \in \mathbb{R}^D : \|\mathbf{x}\|_2 = 1\}.
\tag{A13}
$$

A spherical harmonic $Y_{km}(\cdot)$—where $k$ indexes frequency and $m$ indexes modes of the same frequency—is a harmonic homogeneous polynomial of degree $k$ restricted to the sphere $\mathbb{S}^{D-1}$. For each frequency $k \in \mathbb{Z}$, there are $N(D, k)$ linearly independent polynomials, where

$$N(D, k) = \frac{2k + D - 2}{k} \binom{k+D-3}{k-1}. \tag{A14}$$

## 3.1 Decomposition of the kernel and target function

We remind the reader here of the setting for our theory:

1. Ridge regression using random features with a dot-product limiting kernel.
2. Data drawn uniformly from the unit sphere.

Let $\sigma$ be the Lebesgue measure on $\mathbb{S}^{D-1}$. We will denote the surface area of the sphere as

$$\left| \mathbb{S}^{D-1} \right| = \int_{\mathbb{S}^{D-1}} \mathrm{d}\sigma = \frac{2\pi^{D/2}}{\Gamma(D/2)}. \tag{A15}$$

On the other hand, the uniform probability measure on the sphere, denoted by $\bar{\sigma}$, must integrate to 1, so $\bar{\sigma} = \sigma/|\mathbb{S}^{D-1}|$. Finally, we define the space of real-valued square integrable functions $L^2(\sigma)$ as the Hilbert space with inner product

$$\langle f, g \rangle_{L^2(\sigma)} = \int_{\mathbb{S}^{D-1}} f(\mathbf{x})g(\mathbf{x}) \, \mathrm{d}\sigma(\mathbf{x}) \tag{A16}$$

and $\|f\|_{L^2(\sigma)} = \langle f, f \rangle_{L^2(\sigma)}^{1/2}$. The space $L^2(\bar{\sigma})$ is defined analogously.

Eigendecompositions describe the action of linear operators, not functions, thus we must associate a linear operator with our kernel for its eigenvalues to make sense. The kernel eigenvalues $\lambda_\alpha$ that we will use to compute the error are the eigenvalues of the integral operator $\mathcal{T}_K : L^2(\bar{\sigma}) \to L^2(\bar{\sigma})$ defined as

$$(\mathcal{T}_K f)(\mathbf{x}) = \langle K(\mathbf{x}, \cdot), f(\cdot) \rangle_{L^2(\bar{\sigma})} = \int_{\mathbb{S}^{D-1}} K(\mathbf{x}, \mathbf{x}')f(\mathbf{x}')\mathrm{d}\bar{\sigma}(\mathbf{x}'). \tag{A17}$$

This is because $\bar{\sigma}$ is the data distribution, and these eigenvalues are approximated by the eigenvalues of the kernel matrix evaluated on a large but finite dataset (***Koltchinskii et al., 2000***). Similarly, we define the analogous operator $\mathcal{U}_K : L^2(\sigma) \to L^2(\sigma)$ under the measure $\sigma$ with eigenvalues $\xi_\alpha$. Since $\mathcal{T}_K = \mathcal{U}_K/|\mathbb{S}^{D-1}|$, the eigenvalues are related by

$$\lambda_\alpha = \frac{\xi_\alpha}{|\mathbb{S}^{D-1}|}, \tag{A18}$$

and they share the same eigenfunctions, up to normalization. For the rest of this section we will study eigendecompositions of operator $\mathcal{U}_K$, which may be translated into statements about $\mathcal{T}_K$ via (18) (These differences are liable to cause some confusion and pain when reading the literature).

Under mild technical conditions that our kernels satisfy, Mercer's theorem states that positive semidefinite kernels can be expanded as a series in the orthonormal basis of eigenfunctions $\psi_\alpha$ weighted by nonnegative eigenvalues $\xi_\alpha$:

$$K(\mathbf{x}, \mathbf{x}') = \sum_\alpha \xi_\alpha \psi_\alpha(\mathbf{x})\psi_\alpha(\mathbf{x}'). \tag{A19}$$

Again, $(\lambda_\alpha, \psi_\alpha)$ are eigenpairs for the operator $\mathcal{U}_K$ and form an orthonormal set under the $L^2(\sigma)$ inner product.

As stated earlier, the kernel (***Equation A5***) is spherically symmetric and thus a dot-product kernel. Because of this, we can take the eigenfunctions $\psi_\alpha$ to be the spherical harmonics $Y_{km}$. The index $\alpha$ is a multi-index into mode $m$ of frequency $k$. Writing the Mercer decomposition in the spherical harmonic basis gives:

$$K(\mathbf{x}, \mathbf{x}') = \sum_{k=0}^{\infty} \xi_k \sum_{m=1}^{N(D,k)} Y_{km}(\mathbf{x}) Y_{km}(\mathbf{x}'). \tag{A20}$$

Because our kernel is rotation invariant, all $N(D, k)$ harmonics of frequency $k$ share eigenvalue $\xi_k$. Any function in $L^2(\sigma)$ can be expanded in the spherical harmonic basis as follows:

$$f(\mathbf{x}) = \sum_{k=0}^{\infty} \sum_{m=1}^{N(D,k)} c_{km} Y_{km}(\mathbf{x}), \text{ with } c_{km} = \langle f, Y_{km} \rangle_{L^2(\sigma)}. \tag{A21}$$

The expansion is analogous to that of the Fourier series. In fact when $D = 2$, the spherical harmonics are sines and cosines on the unit circle.

## 3.2 Ultraspherical polynomials

Adding together all harmonics of a given frequency relates them to a polynomial in $t$ by the addition formula

$$\sum_{m=1}^{N(D,k)} Y_{km}(\mathbf{x}) Y_{km}(\mathbf{x}') = \frac{N(D,k)}{|\mathbb{S}^{D-1}|} P_{k,D}(\mathbf{x}^T \mathbf{x}'). \tag{A22}$$

The polynomial $P_{k,D}(t)$ is the $k$ th ultraspherical polynomial. These are also called Legendre or Gegenbauer polynomials, although these usually have different normalizations and can be defined more generally.

The ultraspherical polynomials $\{P_{k,D}\}$ form an orthogonal basis for

$$L^2([-1, 1], (1 - t^2)^{(D-3)/2} \mathrm{d}t).$$

As special cases, $P_{k,2}(t)$ and $P_{k,3}(t)$ are the classical Chebyshev and Legendre polynomials, respectively. For any $D$, the first two of these polynomials are $P_0(t) = 1$ and $P_1(t) = t$. We use the Rodrigues formula (*Atkinson and Han, 2012*), which holds for $k \geq 0$ and $D \geq 2$, to generate these polynomials:

$$P_{k,D}(t) = (-1/2)^k \frac{\Gamma((D-1)/2)}{\Gamma(k + (D-1)/2)} (1 - t^2)^{(3-D)/2} \left( \frac{\phantom{-}}{t} \right)^k (1 - t^2)^{k + (D-3)/2}. \tag{A23}$$

Combining *Equation A20* with the addition formula (*Equation A22*), we can express the kernel in terms of ultraspherical polynomials evaluated at the dot-product of the inputs:

$$K(t) = \sum_{k=0}^{\infty} \xi_k \frac{N(D,k)}{|\mathbb{S}^{D-1}|} P_{k,D}(t). \tag{A24}$$

## 3.3 Computing kernel eigenvalues

The Funk-Hecke theorem states that

$$\int_{\mathbf{x} \in \mathbb{S}^{D-1}} K(\mathbf{x}^T \mathbf{x}') Y_k(\mathbf{x}') \mathrm{d}\sigma(\mathbf{x}') = |\mathbb{S}^{D-2}| Y_k(\mathbf{x}) \int_{-1}^{1} K(t) P_{k,D}(t) (1 - t^2)^{(D-3)/2} \mathrm{d}t. \tag{A25}$$

*Equation A25* implies that the eigenvalues of $\mathcal{U}_K$ are given as

$$\xi_k = |\mathbb{S}^{D-2}| \int_{-1}^{1} K(t) P_{k,D}(t) (1 - t^2)^{(D-3)/2} \mathrm{d}t. \tag{A26}$$

For our kernels, the kernel eigenvalues can be conveniently computed using polar coordinates. When the entries of $\mathbf{J}_1$ are i.i.d. unit Gaussian,

$$
\begin{aligned}
K(t) &= \int_{\mathbb{R}^D} \phi(\mathbf{J}_1^T \mathbf{x} - \theta) \phi(\mathbf{J}_1^T \mathbf{x}' - \theta) (2\pi)^{-D/2} e^{-\|\mathbf{J}_1\|^2/2} \, \mathrm{d}(\mathbf{J}_1)_1 \cdots \mathrm{d}(\mathbf{J}_1)_D \\
&= (2\pi)^{-D/2} \int_0^{\infty} e^{-r^2/2} r^{D-1} \int_{\mathbb{S}^{D-1}} \phi(r\hat{\mathbf{J}}^T \mathbf{x} - \theta) \phi(r\hat{\mathbf{J}}^T \mathbf{x}' - \theta) \mathrm{d}\sigma(\hat{\mathbf{J}}) \mathrm{d}r,
\end{aligned}
$$

where $\hat{\mathbf{x}} = \mathbf{J}_1/r$ and $r = \|\mathbf{J}_1\|$. The ReLU nonlinearity is positively homogeneous, so $\phi(r\hat{\mathbf{J}}^T\mathbf{x} - \theta) = r\phi(\hat{\mathbf{J}}^T\mathbf{x} - \theta/r)$. We can write

$$K(t) = (2\pi)^{-D/2}\int_0^\infty e^{-r^2/2}r^{D+1}\underbrace{\int_{\mathbb{S}^{D-1}}(\hat{\mathbf{J}}^T\mathbf{x} - \theta/r)_+(\hat{\mathbf{J}}^T\mathbf{x}' - \theta/r)_+\mathrm{d}\sigma(\hat{\mathbf{J}})}_{:=K_{\mathrm{shell}}(t;\theta/r)}\mathrm{d}r \tag{A27}$$

$$= (2\pi)^{-D/2}\int_0^\infty e^{-r^2/2}r^{D+1}K_{\mathrm{shell}}(t;\theta/r)\mathrm{d}r,$$

where we have introduced a new kernel $K_{\mathrm{shell}}(t;\theta)$ which is $|\mathbb{S}^{D-1}|$ times the dot-product kernel that arises when the weights are distributed uniformly on the sphere ($\sigma$ is not the probability measure). The above equation shows that the network restricted to inputs $\mathbf{x}, \mathbf{x}' \in \mathbb{S}^{D-1}$ has different kernels depending on whether the weights are sampled according to a Gaussian distribution or uniformly on the sphere. Without the threshold, this difference disappears due to the positive homogeneity of the ReLU (**Churchland et al., 2010**).

Next we expand the nonlinearity in the spherical harmonic basis (following **Bietti and Bach, 2021**; **Bach, 2017**)

$$\phi(\hat{\mathbf{J}}^T\mathbf{x} - \theta) = (\hat{\mathbf{J}}^T\mathbf{x} - \theta)_+ = \sum_{k=0}^\infty a_k(\theta)\sum_{j=1}^{N(D,k)} Y_{kj}(\hat{\mathbf{J}})Y_{kj}(\mathbf{x}), \tag{A28}$$

where the $k$ th coefficient is given by the Funk-Hecke formula (**Equation A25**) as

$$a_k(\theta) = |\mathbb{S}^{D-2}|\int_{-1}^1 (t - \theta)_+ P_k(t)(1 - t^2)^{(D-3)/2}\mathrm{d}t, \tag{A29}$$

and we explicitly note the dependence on $\theta$. Using the representation **Equation A28**, we can recover the eigendecomposition:

$$K_{\mathrm{shell}}(t;\theta) = \int_{\mathbb{S}^{D-1}}(\hat{\mathbf{J}}^T\mathbf{x} - \theta)_+(\hat{\mathbf{J}}^T\mathbf{x}' - \theta)_+\mathrm{d}\sigma(\hat{\mathbf{J}})$$

$$= \sum_{k,k'} a_k(\theta)a_{k'}(\theta)\sum_{j,j'} Y_{kj}(\mathbf{x})Y_{k'j'}(\mathbf{x}')\underbrace{\int_{\mathbb{S}^{D-1}} Y_{kj}(\hat{\mathbf{J}})Y_{k'j'}(\hat{\mathbf{J}})\mathrm{d}\sigma(\hat{\mathbf{J}})}_{\delta_{kk'}\delta_{jj'}} \tag{A30}$$

$$= \sum_k a_k(\theta)^2\sum_j Y_{kj}(\mathbf{x})Y_{kj}(\mathbf{x}')$$

$$= \sum_k a_k(\theta)^2\frac{N(k,D)}{|\mathbb{S}^{D-1}|}P_k(t),$$

which follows from orthonormality and the addition formula (**Equation A22**). We have that $a_k(\theta)^2$ is the $k$ th eigenvalue of $K_{\mathrm{shell}}(t;\theta)$.

Using **Equation A30** in **Equation A27** leads to

$$K(t) = \sum_k \frac{N(k,D)}{|\mathbb{S}^{D-1}|}P_k(t)(2\pi)^{-D/2}\int_0^\infty e^{-r^2/2}r^{D+1}a_k(\theta/r)^2\mathrm{d}r, \tag{A31}$$

i.e. the eigenvalues satisfy

$$\xi_k = (2\pi)^{-D/2}\int_0^\infty e^{-r^2/2}r^{D+1}a_k(\theta/r)^2\mathrm{d}r. \tag{A32}$$

### 3.3.1 Eigenvalues of $K_{\mathrm{shell}}$

It is possible to compute $a_k(\theta)$ analytically (**Bietti and Bach, 2021**; **Bach, 2017**). Letting

$$I_{\alpha,k}(\theta) = \int_\theta^1 t^\alpha P_k(t)(1 - t^2)^{(D-3)/2}\mathrm{d}t, \tag{A33}$$

we have that **Equation A29** reduces to $a_k(\theta) = |\mathbb{S}^{D-2}|\left(I_{1,k}(\theta^*) - \theta I_{0,k}(\theta^*)\right)$. **Equation A33** requires $\theta \in [-1, 1]$, but $\theta/r \to \pm\infty$ in **Equation A32** as $r \to 0$. So we take $\theta^* = \min(\max(\theta, -1), 1)$, which still assures that **Equation A29** is satisfied. For the rest of this section, assume wlog that $\theta \in [-1, 1)$.

Using Rodrigues' formula (**Equation A23**) in **Equation A33** gives

$$I_{\alpha,k}(\theta) \;=\; \underbrace{(-1/2)^k \frac{\Gamma((D-1)/2)}{\Gamma(k+(D-1)/2)}}_{:=C} \int_\theta^1 t^\alpha \left(\frac{d}{dt}\right)^k (1-t^2)^{k+(D-3)/2} dt$$

$$= C \int_\theta^1 t^\alpha \left(\frac{d}{dt}\right)^k (1-t^2)^{k+(D-3)/2} dt$$

which may be integrated by parts. We will treat $\alpha = 0$ and $1$ separately.

In the case of $\alpha = 0$, since $t^\alpha = 1$ we have the integral of a derivative, so for $k \geq 1$

$$I_{0,k}(\theta) \;=\; C \int_\theta^1 \left(\frac{d}{dt}\right)^k (1-t^2)^{k+(D-3)/2} dt$$

$$= C \left(\frac{d}{dt}\right)^{k-1} (1-t^2)^{k+(D-3)/2} \Big|_\theta^1$$

$$= -C \left(\frac{d}{dt}\right)^{k-1} (1-t^2)^{k+(D-3)/2} \Big|_{t=\theta} \qquad (k \geq 1)$$

When $k = 0$ we find that

$$I_{0,0}(\theta) \;=\; \int_\theta^1 (1-t^2)^{(D-3)/2} dt$$

$$= t \,_2F_1(1/2, (3-D)/2; 3/2; t^2) \Big|_\theta^1$$

$$= \frac{\sqrt{\pi}\,\Gamma((D-1)/2)}{2\Gamma(D/2)} - \theta \,_2F_1(1/2, (3-D)/2; 3/2; \theta^2).$$

For $\alpha = 1$, we integrate by parts once and find that for $k \geq 2$,

$$I_{1,k}(\theta) \;=\; C \int_\theta^1 t \left(\frac{d}{dt}\right)^k (1-t^2)^{k+(D-3)/2} dt$$

$$= C \left[ t \left(\frac{d}{dt}\right)^{k-1} (1-t^2)^{k+(D-3)/2} \Big|_\theta^1 - \int_\theta^1 \left(\frac{d}{dt}\right)^{k-1} (1-t^2)^{k+(D-3)/2} dt \right]$$

$$= C \left[ t \left(\frac{d}{dt}\right)^{k-1} (1-t^2)^{k+(D-3)/2} \Big|_\theta^1 - \left(\frac{d}{dt}\right)^{k-2} (1-t^2)^{k+(D-3)/2} \Big|_\theta^1 \right]$$

$$= C \left[ \left(\frac{d}{dt}\right)^{k-2} (1-t^2)^{k+(D-3)/2} - t \left(\frac{d}{dt}\right)^{k-1} (1-t^2)^{k+(D-3)/2} \right] \Big|_{t=\theta} \qquad (k \geq 2)$$

When $\alpha = 0$, we have a straightforward integral

$$I_{1,0}(\theta) \;=\; \int_\theta^1 t(1-t^2)^{(D-3)/2} dt$$

$$= \frac{(1-\theta^2)^{(D-1)/2}}{(D-1)} = I_{0,1}(\theta).$$

Finally, for $k = 1$, we obtain

$$I_{1,1}(\theta) \;=\; \int_\theta^1 t^2 (1-t^2)^{(D-3)/2} dt$$

$$= (t^3/3) \,_2F_1(3/2, (3-D)/2; 5/2; t^2) \Big|_\theta^1$$

$$= \frac{\sqrt{\pi}\,\Gamma((D-1)/2)}{4\Gamma((D+2)/2)} - (\theta^3/3) \,_2F_1(3/2, (3-D)/2; 5/2; \theta^2)$$

### 3.3.2 Properties of the eigenvalues of $K_{\text{shell}}$

The above show that for $k \geq 2$

$$a_k = |\mathbb{S}^{D-2}|(I_{1,k}(\theta) - \theta I_{0,k}(\theta)) \propto \left(\frac{d}{dt}\right)^{k-2} (1-t^2)^{k+(D-3)/2}\Bigg|_{t=\theta}. \tag{A34}$$

Taking $\theta = -1$ leads to $a_k = 0$, since fewer derivatives than $k + (D-3)/2$ appear in **Equation A34**, which reflects the fact that higher degree ultraspherical polynomials are orthogonal to a linear function. Furthermore, since $1 - t^2$ is an even function, the parity of $a_k$ as a function of $\theta$ matches the parity of $k$. However, $a_k$ appears squared in **Equation A32**, so $\xi_k$ will always be an even function of $\theta$. This explains the parity symmetry of the eigenvalues with coding level for $k \geq 2$. Also, **Equation A34** for $\theta = 0$ gives $a_k = 0$ when $k$ is odd, as was shown by **Bach, 2017**; **Basri et al., 2019**. This is because

$$\left(\frac{d}{dt}\right)^p (1-t^2)^{p+\ell}\Bigg|_{t=0} = \left(\frac{d}{dt}\right)^p (1-t)^{p+l}(1+t)^{p+l}\Bigg|_{t=0}$$

$$= \sum_{j=0}^p \binom{p}{j}\left(\left(\frac{d}{dt}\right)^j (1-t)^{p+l}\right)\left(\left(\frac{d}{dt}\right)^{p-j} (1+t)^{p+l}\right)\Bigg|_{t=0}$$

$$= \sum_{j=0}^p \binom{p}{j}(-1)^j\left(\left(\frac{d}{dt}\right)^j (1+t)^{p+l}\right)\left(\left(\frac{d}{dt}\right)^{p-j} (1+t)^{p+l}\right)\Bigg|_{t=0}$$

$$= 0 \text{ if } p \text{ is odd,}$$

because the $j$ and $p - j$ terms have opposite parity and cancel.

We may also compute the tail asymptotics of these eigenvalues for large $k$. Let $p = k - 2$ and $\ell = (D+1)/2$, so we want to evaluate

$$\left(\frac{d}{dt}\right)^p (1-t^2)^{p+\ell} = \frac{p!}{2\pi i}\oint \frac{(1-z^2)^{p+\ell}}{(z-t)^{p+1}} dz$$

$$= \frac{p!}{2\pi i}\oint e^{(p+o(p))F(z)} dz$$

for large $p$ at $t = \theta \in (-1, 1)$. The first line follows from Cauchy's intergral formula for a counterclockwise contour encircling $t$, and the second comes from defining

$$F(z) := \log(1 - z^2) - \log(z - t) \sim (1 + \ell/p)\log(1 - z^2) - (1 + 1/p)\log(z - t),$$

when $p$ is large and $\ell$ is constant. We will use the saddle point method (**Butler, 2007**) to evaluate the contour integral asymptotically, ignoring the $o(p)$ term in the exponent. Note that the only singularity in the original integrand occurs at $z = t$.

The function $F$ has derivatives

$$F'(z) = \frac{-2z}{1-z^2} - \frac{1}{z-t},$$

$$F''(z) = \frac{1}{(z-t)^2} - \frac{4z^2}{(1-z^2)^2} - \frac{2}{1-z^2}.$$

We find the saddle points by setting $F'(z) = 0$. This leads to a quadratic equation with two roots: $z_\pm = t \pm \sqrt{t^2 - 1} = \text{sgn}(t)(|t| \pm i\sqrt{1 - t^2})$. Since these are evaluated at $t = \theta$ with $|\theta| < 1$, both roots are complex, $|z_\pm| = 1$, and $F''(z_\pm) \neq 0$. Also, the saddle points avoid the singularity in the original integrand, so we can deform our contour to pass through these points and apply the standard approximation.

Applying the saddle point approximation, we obtain

$$\left(\frac{d}{dt}\right)^p (1-t^2)^{p+\ell} \simeq \frac{p!}{2\pi i} \oint e^{pF(z)} dz$$

$$\simeq \frac{p!}{2\pi i} \sum_{z_0 \in \{z_+, z_-\}} e^{pF(z_0)} e^{i(\pi - \arg F''(z_0))/2} \left(\frac{2\pi}{p|F''(z_0)|}\right)^{1/2}$$

$$\leq cp! \sum_{z_0 \in \{z_+, z_-\}} e^{pF(z_0)} p^{-1/2}$$

$$= cp! \, p^{-1/2} \left(\left(\frac{1-z_+^2}{z_+ - t}\right)^p + \left(\frac{1-z_-^2}{z_- - t}\right)^p\right)$$

$$= cp! \, p^{-1/2} \left((-2z_+)^p + (-2z_-)^p\right)$$

$$\leq 2cp! \, p^{-1/2} (-2)^p$$

for some $c$ which is constant in $p$ and depends on $D$. In the last step, we use that $z_+^p + z_-^p \leq 2$ since $z_\pm$ are conjugate pairs with magnitude 1.

Now recall the full equation for the coefficients:

$$a_k = |\mathbb{S}^{D-2}|(-1/2)^k \frac{\Gamma((D-1)/2)}{\Gamma(k+(D-1)/2)} \left(\frac{d}{dt}\right)^p (1-t^2)^{p+\ell}.$$

Plugging in the result from the saddle point approximation, substituting $p = k - 2$, and dropping all terms that are constant in $k$, we find that

$$a_k \leq C'(-1/2)^k \frac{(k-2)! \, k^{-1/2} (-2)^k}{\Gamma(k+(D-1)/2)}$$

$$= C' k^{-1/2} \frac{\Gamma(k-1)}{\Gamma(k+(D-1)/2)}$$

$$= C' k^{-D/2 - 1},$$

where $C'$ is a new constant. The rate of $k^{-D/2-1}$ is the same decay rate found by *Bach, 2017*; *Bietti and Bach, 2021* using a different mathematical technique for $\theta = 0$. These decay rates are important for obtaining general worst-case bounds for kernel learning of general targets; (*Bach, 2012*) is an example.

## 3.4 Gaussian process targets

Taking our target function to be a GP on the unit sphere $f(\mathbf{x}) \sim \mathbf{GP}(0, C)$ with some covariance function $C : \mathbb{S}^{D-1} \times \mathbb{S}^{D-1} \to \mathbb{R}$, we can represent our target function by performing an eigendecomposition of the covariance operator $\mathcal{U}_C$. When $C$ itself is spherically symmetric and positive definite, this becomes

$$C(\mathbf{x}^\mu, \mathbf{x}^\nu) = \sum_{k=0}^{\infty} \rho_k \sum_{m=1}^{N(D,k)} Y_{km}(\mathbf{x}^\mu) Y_{km}(\mathbf{x}^\nu), \tag{A35}$$

where $\rho_k > 0$ are the eigenvalues. Then a sample from the GP with this covariance function is a random series

$$f(\mathbf{x}) = \sum_{k=0}^{\infty} \sqrt{\rho_k} \sum_{m=1}^{N(D,k)} g_{km} Y_{km}(\mathbf{x}), \tag{A36}$$

where $g_{km} \sim \mathcal{N}(0, 1)$ by the Kosambi-Karhunen–Loève theorem (*Kosambi, 1943*). In other words, the coefficient of $Y_{km}$ in the series expansion of $f(\mathbf{x})$ is $c_{km} = \sqrt{\rho_k} g_{km}$.

We take the squared exponential covariance on the sphere

$$C(\mathbf{x}^\mu, \mathbf{x}^\nu) = \exp\left(\frac{-\|\mathbf{x}^\mu - \mathbf{x}^\nu\|^2}{2\gamma^2}\right) = \exp\left(\frac{t-1}{\gamma^2}\right), \tag{A37}$$

for $t = \mathbf{x}^\mu \cdot \mathbf{x}^\nu$ and length scale $\gamma$.

## 3.5 Numerical details

All of our spherical harmonic expansions are truncated at frequency $N_k$. This is typically $N_k = 50$ for experiments in $D = 3$ dimensions. In higher dimensions, $N(D, k)$ grows very quickly in $k$, requiring truncation at a lower frequency.

To compute the kernel eigenvalues $\lambda_k$, we can either numerically integrate the Funk-Hecke formula (*Equation A25*) or compute the coefficients $a_k(\theta/r)$ semi-analytically, following *Equation A34*, then integrate *Equation A32* with numerical quadrature and rescale by *Equation A18*.

We use the Funk-Hecke formula (*Equation A25*) and numerical quadrature to find $\rho_k$. To compute the expected error using *Equation A38*, we use $\mathbb{E}[c_\alpha^2] = \rho_k$. After generating a sample from the GP, we normalize the functions by dividing the labels and coefficients by their standard deviation. This ensures that the relative mean squared error is equivalent to the mean squared error computed in the next section.

## 4 Calculation of generalization error

The generalization error of kernel ridge regression is derived in *Canatar et al., 2021a*; *Simon et al., 2021*; *Gerace et al., 2021*, which show that the mean squared error, in the absence of noise in the target, can be written as

$$\mathbb{E}_{\mathbf{x}} \, (f(\mathbf{x}) - \hat{f}(\mathbf{x}))^2 = \sum_\alpha \beta_\alpha c_\alpha^2, \tag{A38}$$

where $\beta_\alpha$ depend on $P$ and the kernel but not on the target, and $c_\alpha$ are the coefficients (*Equation A21*) of the target function in the basis $L^2(\sigma)$. The exact form of this expression differs from that given in *Canatar et al., 2021b* due to differences in the conventions we take for our basis expansions. Specifically,

$$\beta_\alpha = \left(\frac{1}{1 - \chi}\right) \left(\frac{\kappa}{\lambda_\alpha P + \kappa}\right)^2, \tag{A39}$$

where $\alpha$ indexes the kernel eigenfunctions and

$$\chi = \sum_\alpha \frac{\lambda_\alpha^2 P}{(\lambda_\alpha P + \kappa)^2}, \tag{A40}$$

$$\kappa = \alpha_{\text{ridge}} + \sum_\alpha \frac{\lambda_\alpha \kappa}{\lambda_\alpha P + \kappa}, \tag{A41}$$

with $\alpha_{\text{ridge}}$ the ridge parameter. Note that *Equation A41* is an implicit equation for $\kappa$, which we solve by numerical root-finding.

Thus,

$$\mathbb{E}_{\mathbf{x}} \, (f(\mathbf{x}) - \hat{f}(\mathbf{x}))^2 = C_1 \sum_\alpha \left(\frac{c_\alpha}{\lambda_\alpha + C_2}\right)^2, \tag{A42}$$

with $C_1 = \left(\frac{1}{1-\chi}\right) \frac{\kappa^2}{P^2}$ and $C_2 = \frac{\kappa}{P}$.

## 5 Dense-sparse networks

To compare with more realistic networks, we break the simplifying assumption that $\mathbf{J}^{\text{eff}}$ is densely connected and instead consider sparse connections between the input and expansion layer. Consider a random matrix $\mathbf{J}^{\text{eff}} = \mathbf{J}\mathbf{A}$, where $\mathbf{A} \in \mathbb{R}^{N \times D}$ and $\mathbf{J} \in \mathbb{R}^{M \times N}$, with $N > D$ and $M > N$. The entries of $\mathbf{A}$ are i.i.d. Gaussian, i.e. $A_{ij} \sim \mathcal{N}\left(0, 1/D\right)$. In contrast, $\mathbf{J}$ is a sparse matrix with *exactly K nonzero entries per row*, and nonzero entries equal to $1/\sqrt{K}$. With these scaling choices, the elements of $\mathbf{J}^{\text{eff}}$ are of order $1/\sqrt{D}$, which is appropriate when the input features $x_i$ are order 1. This is in contrast to the rest of this paper, where we considered features of order $1/\sqrt{D}$ and therefore assumed order 1 weights. The current scaling allows us to study the properties of $\mathbf{J}^{\text{eff}}$ for different values of $D$, $N$ and $K$.

First, we examine properties of $\mathbf{J}^{\text{eff}} = \mathbf{J}\mathbf{A}$ under these assumptions. Recall that the rows of $\mathbf{J}^{\text{eff}}$ are the weights of each hidden layer neuron. Since $\mathbf{J}^{\text{eff}}$ is Gaussian, any given row $\mathbf{J}_i^{\text{eff}} \in \mathbb{R}^D$ is marginally Gaussian and distributed identically to any other row. But the rows are *not* independent, since they

are all linear combinations of the rows of $\mathbf{A}$. Thus, the kernel limit of an infinitely large dense-sparse network is equal to that of a fully dense network, but convergence to that kernel behaves differently and requires taking a limit of both $N, M \to \infty$. In this section, we study how finite $N$ introduces extra correlations among the rows of $\mathbf{J}^{\text{eff}}$ compared to dense networks.

The distribution of $\mathbf{J}^{\text{eff}}$ is spherically symmetric in the sense that $\mathbf{J}^{\text{eff}}$ and $\mathbf{J}^{\text{eff}}\mathbf{Q}$ have the same distribution for any rotation matrix $\mathbf{Q} \in \mathbb{R}^{D \times D}$. In contrast, a densely connected network with weights $\mathbf{G}$ drawn i.i.d. as $G_{ij} \sim \mathcal{N}(0, 1/D)$ will of course have independent rows and also be spherically symmetric. The spherical Gaussian is the only vector random variable which is spherically symmetric with independent entries (***Nash and Klamkin, 1976***). Furthermore, each row of $\mathbf{G}$ may be rotated by a *different* orthogonal matrix and the resulting random variable would still have the same distribution.

With these symmetry considerations in mind, the statistics of the rows of $\mathbf{J}^{\text{eff}}$ can be described by their multi-point correlations. The simplest of these is the two-point correlation, which in the case of spherical symmetry is captured by the overlaps:

$$\nu_{ij} := \sum_{k=1}^{D} (\mathbf{J}^{\text{eff}})_{ik}(\mathbf{J}^{\text{eff}})_{jk} = \sum_{k=1}^{D} \sum_{m,n=1}^{N} J_{in}A_{nk}J_{jm}A_{mk} \ . \tag{A43}$$

The overlap $\nu_{ij}$ is doubly stochastic: one source of stochasticity are the elements of $\mathbf{A}$, and the second one is the random sampling of nonzero elements of $\mathbf{J}$. Ideally, we are interested in studying the statistics of $\nu_{ij}$ when varying $i$ and $j$, i.e. when $\mathbf{J}$ varies (since the rows of $\mathbf{J}$ are sampled independently from each other). However, this will leave us with the quenched disorder given by the specific realization of $\mathbf{A}$. To obtain a more general and interpretable result, we want to compute the probability distribution

$$P_{\mathbf{A},\mathbf{J}}\left(\nu_{ij}\right) = \underset{\mathbf{A}}{\mathbb{E}}\left[\underset{\mathbf{J}}{\mathbb{E}}\left[\delta\left(\nu_{ij} - \sum_{k=1}^{D}\sum_{m,n=1}^{N} J_{in}J_{jm}A_{nk}A_{mk}\right)\right]\right] \quad . \tag{A44}$$

Notice that the order in which we perform the averaging is irrelevant.

## 5.1 Computation of the moment-generating function

Instead of computing directly the probability distribution in ***Equation A44***, we compute the moment-generating function

$$Z(\mu) := \underset{\mathbf{A}}{\mathbb{E}}\left[\underset{\mathbf{J}}{\mathbb{E}}\left[\exp\left(\mu\nu_{ij}\right)\right]\right] , \tag{A45}$$

which fully characterizes the probability distribution of $\nu_{ij}$. We indicate the set of indices in which the $i$-th row of $\mathbf{J}$ takes nonzero values by $S^i = \{S_1^i, S_2^i, \dots, S_K^i\}$ such that $J_{iS_l^i} \neq 0$, $\forall l = 1, \dots, K$, and analogously for the $j$-th row. We also indicate the intersection $S^{ij} = S^i \cap S^j$, i.e. the set of indices in which *both* the $i$-th and the $j$-th rows are nonzero. $S^{ij}$ has size $0 \leq |S^{ij}| \leq K$. Notice that setting $i = j$ causes $|S^{ij}| = K$ deterministically. With this definitions, the overlap can be written as

$$\nu_{ij} = \sum_{k=1}^{D} \sum_{m \in S^i} \sum_{n \in S^j} J_{in}J_{jm}A_{nk}A_{mk} = \frac{1}{K}\sum_{k=1}^{D} \sum_{m \in S^i} \sum_{n \in S^j} A_{nk}A_{mk} \tag{A46}$$

We start by perform swapping the averaging order in ***Equation A45*** and averaging over $\mathbf{A}$.

$$\begin{aligned} Z(\mu) = \ & \underset{\mathbf{J}}{\mathbb{E}}\left[\int \left(\prod_{m=1}^{N}\prod_{l=1}^{D} \mathcal{D}A_{ml}\right) \exp\left(\frac{\mu}{K}\sum_{k=1}^{D}\sum_{m \in S^i}\sum_{n \in S^j} A_{nk}A_{mk}\right)\right] \\ = \ & \underset{\mathbf{J}}{\mathbb{E}}\left[\int \left(\prod_{m \in S^i \cup S^j}\prod_{l=1}^{D} \mathcal{D}A_{ml}\right) \exp\left(\frac{\mu}{K}\sum_{k=1}^{D}\sum_{m \in S^i}\sum_{n \in S^j} A_{nk}A_{mk}\right)\right] \\ = \ & \underset{\mathbf{J}}{\mathbb{E}}\left[\prod_{k=1}^{D}\int \left(\prod_{m \in S^i \cup S^j} \mathcal{D}A_{mk}\right) \exp\left(\frac{\mu}{K}\sum_{m \in S^i}\sum_{n \in S^j} A_{nk}A_{mk}\right)\right] , \end{aligned}$$

where in the first equality we marginalized over all the elements of $\mathbf{A}$ which do not enter the definition of $\nu_{ij}$, i.e. we went from having to integrate over $N \times D$ variables to only $|S^i \cup S^j| \times D = (2K - |S^{ij}|) \times D$ variables. In the second equality we factorized the columns of $\mathbf{A}$.

We now explicitly compute integral for a fixed value of $k$, by reducing it to a Gaussian integral:

$$\int \left( \prod_{m \in S^i \cup S^j} \mathcal{D}A_{mk} \right) \exp\left( \frac{\mu}{K} \sum_{m \in S^i} \sum_{n \in S^j} A_{nk}A_{mk} \right)$$

$$= \int \left( \prod_{m \in S^i \cup S^j} dA_{mk} \right) (2\pi D)^{-\frac{|S^i \cup S^j|}{2}} \exp\left( \frac{\mu}{K} \sum_{m \in S^i} \sum_{n \in S^j} A_{nk}A_{mk} - \frac{D}{2} \sum_{r \in S^i \cup S^j} A_{rk}^2 \right)$$

$$= \int \left( \prod_{m \in S^i \cup S^j} dA_{mk} \right) (2\pi D)^{-\frac{|S^i \cup S^j|}{2}} \exp\left( -\frac{D}{2} \sum_{r \in S^i \cup S^j} A_{rk}P_{rs}A_{sk} \right)$$

$$= \det(\mathbf{P})^{-1/2},$$

where $\mathbf{P} \in \mathbb{R}^{|S^i \cup S^j| \times |S^i \cup S^j|}$, which has a 3-by-3 block structure and can be written as

$$\mathbf{P} = \begin{pmatrix} I_{K-|S^{ij}|} & -\frac{\mu}{KD}1_{|S^{ij}|} & -\frac{\mu}{KD}1_{K-|S^{ij}|} \\ -\frac{\mu}{KD}\vec{1}_{|S^{ij}| \times (K-|S^{ij}|)} & I_{|S^{ij}|} - 2\frac{\mu}{KD}1_{|S^{ij}|} & -\frac{\mu}{KD}1_{|S^{ij}| \times (K-|S^{ij}|)} \\ -\frac{\mu}{KD}1_{K-|S^{ij}|} & -\frac{\mu}{KD}1_{|S^{ij}| \times (K-|S^{ij}|)} & I_{K-|S^{ij}|} \end{pmatrix} , \tag{A47}$$

where $I_n$ is the $n$-by-$n$ identity matrix and $\mathbf{J}^{\text{eff}}$ is the $n$-by-$m$ matrix of all ones (if $m$ is omitted, then it is $n$-by-$n$). Due to the block structure, the determinant of the matrix above is identical to the determinant of a 3-by-3 matrix

$$\det(\mathbf{P}) = \det \begin{pmatrix} 1 & -\frac{\mu}{KD}|S^{ij}| & -\frac{\mu}{KD}(K-|S^{ij}|) \\ -\frac{\mu}{KD}(K-|S^{ij}|) & 1 - 2\frac{\mu}{KD}|S^{ij}| & -\frac{\mu}{KD}(K-|S^{ij}|) \\ -\frac{\mu}{KD}(K-|S^{ij}|) & -\frac{\mu}{KD}|S^{ij}| & 1 \end{pmatrix}$$

$$= \frac{K^2D^2 - K^2\mu^2 + |S^{ij}|^2\mu^2 - 2DK|S^{ij}|\mu}{K^2D^2} . \tag{A48}$$

By plugging this result into the expression for the moment-generating function, we have that

$$Z(\mu) = \mathop{\mathbb{E}}_{\mathbf{J}} \left[ \left( \frac{K^2D^2 - K^2\mu^2 + |S^{ij}|^2\mu^2 - 2DK|S^{ij}|\mu}{K^2D^2} \right)^{-D/2} \right] . \tag{A49}$$

This expression is our core result, and needs to be averaged over $\mathbf{J}$. This average can be written explicitly by noticing that, when $i \neq j$, $|S^{ij}|$ is a random variable that follows a hypergeometric distribution in which the number of draws is equal to number of success state and is equal to $K$. By using the explicit expression of the probability mass function of a hypergeometric distribution, we have that

$$Z(\mu) = \sum_{s=0}^{K} \frac{\binom{K}{s}\binom{N-K}{K-s}}{\binom{N}{K}} \left( \frac{K^2D^2 - K^2\mu^2 + s^2\mu^2 - 2DKs\mu}{K^2D^2} \right)^{-D/2} . \tag{A50}$$

Notice that the term $s = 0$ yields the same moment-generating function (up to a factor) as for a fully-connected $\mathbf{J}^{\text{eff}}$ with Gaussian i.i.d. entries with variance $1/D$. In contrast, when $i = j$ we obtain

$$Z_{i=j}(\mu) = \left( 1 - \frac{2\mu}{D} \right)^{-D/2} . \tag{A51}$$

## 5.2 Computation of the moments of $\nu_{ij}$

In this section, we assume that $i \neq j$ and use the moment-generating function to compute the moments of $\nu_{ij}$. The non-central moments of the overlap are easily obtained from the moment-generating function as

$$\mathop{\mathbb{E}}_{\mathbf{J}}\left[ \nu_{ij}^q \right] = \frac{d^q}{d\mu^q} Z(\mu)|_{\mu=0}, \tag{A52}$$

which can be computed in a symbolic manipulation tool.

We now explicitly compute the first two moments of $\nu_{ij}$.

$$\mathbb{E}_{\mathbf{J}}\left[\nu_{ij}\right] = \frac{d}{d\mu}Z(\mu)|_{\mu=0} = \frac{1}{K}\mathbb{E}_{\mathbf{J}}\left[|S^{ij}|\right] = \frac{K}{N} \quad , \tag{A53}$$

where we used the fact that the mean of $s \sim \text{Hypergeom}(N, K, K)$ is given by $\mathbb{E}[s] = \frac{K^2}{N}$. For the second moment, we have

$$\mathbb{E}\left[\nu_{ij}^2\right] = \mathbb{E}_{\mathbf{J}}\left[\frac{K^2 + |S^{ij}|^2(1+D)}{K^2 D}\right] = \frac{1}{D} + \frac{D+1}{D}\left(\frac{(N-K)^2}{N^2(N-1)} + \frac{K^2}{N^2}\right),$$

while to compute the variance we use the law of total variance

$$
\begin{aligned}
\text{Var}(\nu_{ij}) &= \mathbb{E}_{s}[\text{Var}(\nu_{ij}|s)] + \text{Var}(\mathbb{E}\left[\nu_{ij}|s\right]) \\
&= \frac{1}{D} + \frac{D+1}{K^2 D}\mathbb{E}[s^2] - \frac{1}{K^2}\mathbb{E}[s^2] + \text{Var}\left(\frac{1}{K}s\right) \\
&= \frac{1}{D} + \frac{D+1}{D}\left(\frac{(N-K)^2}{N^2(N-1)} + \frac{K^2}{N^2}\right) - \frac{K^2}{N^2} \\
&= \frac{1}{D} + \frac{D+1}{D}\left(\frac{(N-K)^2 + (N-1)K^2 - (N-1)K^2\frac{D}{D+1}}{N^2(N-1)}\right) \\
&= \frac{1}{D} + \frac{D+1}{D}\frac{1}{N-1} + \frac{1}{D}\frac{K^2 - 2(D+1)K}{N(N-1)} + \frac{K^2}{N^2(N-1)}.
\end{aligned}
$$

As $N \to \infty$ we have that $\text{Var}(\nu_{ij}) \sim \frac{1}{D}$, which is the same variance of the overlap for a fully-connected $\mathbf{J}^{\text{eff}}$ with Gaussian i.i.d. entries. This is expected since when $N$ is large, the probability of $i$ and $j$ having common afferents goes to zero.

## 5.3 Comparison to clustered embedding

Instead of distributed embedding, i.e. $\mathbf{A}$ being a Gaussian matrix, here we consider a clustered embedding by setting

$$\mathbf{A} = \mathbf{I}_D \otimes \mathbf{1}_{N/D}. \tag{A54}$$

i.e. the Kronecker product of the $D$-dimensional identity matrix and a vector of all ones and length $N/D$. This means that we can separate the input layer of $N$ neurons in $D$ non overlapping subsets $B_n = \{\frac{N}{D}(n-1)+1, \frac{N}{D}(n-1)+2, \ldots, \frac{N}{D}n\}$, each of size $N/D$, and we can write

$$\mathrm{A}_{mn} = \begin{cases} 1 & \text{if } m \in B_n \\ 0 & \text{otherwise} \end{cases}. \tag{A55}$$

In this case the overlap is given by

$$\nu_{ij} = \frac{1}{K}\sum_{l=1}^{D}\sum_{m\in S^i}\sum_{n\in S^j} 1\left[m \in B_l\right] 1\left[n \in B_l\right], \tag{A56}$$

where $1[\cdot]$ is the indicator function, i.e. it is one if the argument is true and zero if the argument is false. We indicate by $K_l^i$ the number of elements of $S^i$ which belongs to group $l$, i.e. $K_l^i = \sum_{m\in S^i} 1[m \in B_l]$. The overlap can then be written as

$$\nu_{ij} = \frac{1}{K}\sum_{l=1}^{D} K_l^i K_l^j. \tag{A57}$$

The vector $\mathbf{K}^i = (K_1^i, \ldots, K_D^i)$ follows a multivariate hypergeometric distribution with $D$ classes, $K$ draws, a population of size $N$, and number of successes for each class equal to $N/D$. Notice that $\mathbf{K}^i$ and $\mathbf{K}^j$ are independent from each other since each neuron samples its pre-synaptic partners independently. We can now compute explicitly the mean of $\nu_{ij}$ using the fact that $\mathbb{E}\left[K_l^i\right] = \frac{K}{D}$

$$\mathbb{E}\left[\nu_{ij}\right] = \frac{1}{K}\sum_{l=1}^{D}\mathbb{E}\left[K_L^i\right]^2 = \frac{K}{D} \quad . \tag{A58}$$

Similarly, we can write the second moment of $\nu_{ij}$ as

$$\mathbb{E}\left[\nu_{ij}^2\right] = \frac{1}{K^2}\left(\sum_{l=1}^{D}\mathbb{E}\left[(K_l^i)^2\right]^2 + \sum_{l\neq l'}\mathbb{E}\left[K_l^iK_{l'}^i\right]^2\right). \tag{A59}$$

Once again, we can use known result for variance and covariance of multivariate hypergeometric variables to simplify the above expression. Indeed, we can write

$$\mathbb{E}\left[(K_l^i)^2\right] = K\frac{N-K}{N-1}\frac{D-1}{D^2} + \frac{K^2}{D^2} \tag{A60}$$

$$\mathbb{E}\left[K_l^iK_{l'}^i\right] = -K\frac{N-K}{N-1}\left(\frac{1}{D}\right)^2 + \frac{K^2}{D^2} \tag{A61}$$

from which we obtain the final expression for the second moment

$$\mathbb{E}\left[\nu_{ij}^2\right] = \frac{K^2}{D^2} + \frac{1}{D}\left(1-\frac{1}{D}\right)\left(\frac{N-K}{N-1}\right)^2. \tag{A62}$$

