## [Editor Report]

Models of cerebellar function and the coding of inputs in the cerebellum often assume that random stimuli are a reasonable stand-in for real stimuli. However, the important contribution of this paper is that conclusions about optimality and sparseness in these models do not generalize to potentially more realistic sets of stimuli, for example, those drawn from a low-dimensional manifold. The mathematical analyses in the paper are convincing and possible limitations, including the abstraction from biological details, are well discussed.

---

## [Decision Letter]

**Decision letter after peer review:**

Thank you for submitting your article "Task-dependent optimal representations for cerebellar learning" for consideration by *eLife*. Your article has been reviewed by 3 peer reviewers, including Jörn Diedrichsen as Reviewing Editor and Reviewer #1, and the evaluation has been overseen by Michael Frank as the Senior Editor. The following individual involved in the review of your submission has agreed to reveal their identity: Henrik Jörntell (Reviewer #3).

The reviewers have discussed their reviews with one another, and the Reviewing Editor has drafted this to help you prepare a revised submission. You will see that the recommended revisions most concern clarity of presentation, as well as tempering some of the claims.

Essential revisions:

1) All reviewers had questions about specific details of the study (see below comments). I hope you can use these questions as a guideline to improve the clarity and accessibility of the paper. As suggested by reviewer #2, this may involve moving some details from the supplementary materials to the main manuscript.

2) The current *eLife* assessment (see below) emphasizes the main limitations of the paper – namely the question of to what degree the conclusions of the paper would change if more specific (and biologically more plausible) details about the cerebellar circuit would be taken into account. I hope you will take the opportunity to respond in detail to these points in your response to the reviewer document, which will be posted alongside the revised version of the manuscript and cover the main issues in the Discussion section of the main document.*Reviewer #1 (Recommendations for the authors):*

This paper provides compelling and clear analyses that show that the coding level (sparsity) of the granule-cell layer of a cerebellar-like network does not only change the dimensionality of the representation, but also the inductive bias of the network: Higher sparsity biases the network to learning more high-frequency representations. Depending on the dominant frequencies of the target function, different coding levels are therefore optimal. The results are important/fundamental to theories of cerebellar learning and speak to a relevant ongoing debate in cerebellar learning, but will be of interest to readers outside of cerebellar neurophysiology.

I had two problems in understanding the paper that the authors hopefully can clarify in the revision:

Page 8: Third paragraph: At this point in the text it is a bit unclear why the K(x * x') changes shape as shown in Figure 3c. I assume the shape of the kernel K depends on the activation function in the hidden layer. It may be useful for the reader if you could give an example of the Kernel for the specific case you are showing in Figure 3c.

Page 11, 4th paragraph: Why is the distribution corr(J^eff^_i_,J^eff^_j_) uniform on -1 to 1? Since both are high-dimensional random vectors, should the correlation not be centered around 0? The target of uniform distribution needs to be better explained to make this analysis accessible.*Reviewer #2 (Recommendations for the authors):*

Here, a simple model of cerebellar computation is used to study the dependence of task performance on input type: it is demonstrated that task performance and optimal representations are highly dependent on task and stimulus type. This challenges many standard models which use simple random stimuli and concludes that the granular layer is required to provide a sparse representation. This is a useful contribution to our understanding of cerebellar circuits, though, in common with many models of this type, the neural dynamics and circuit architecture are not very specific to the cerebellum, the model includes the feed-forward structure and the high dimension of the granule layer, but little else. This paper has the virtue of including tasks that are more realistic, but by the paper's own admission, the same model can be applied to the electrosensory lateral line lobe and it could, though it is not mentioned in the paper, be applied to the dentate gyrus and large pyramidal cells of CA3. The discussion does not include specific elements related to, for example, the dynamics of the Purkinje cells or the role of Golgi cells, and, in a way, the demonstration that the model can encompass different tasks and stimuli types is an indication of how abstract the model is. Nonetheless, it is useful and interesting to see a generalization of what has become a standard paradigm for discussing cerebellar function.

I was impressed by the clarity of this manuscript. My only comment is that I found too much was deferred to the appendix, I thought the bits and pieces in the appendix were very clarifying, and given the appendix contains short pieces of extra information explaining the exact nature of the models and tasks, the manuscript would have been easier to follow and think about if these had just been integrated into the text. Often you read papers and wish some details had been shifted into an appendix since they distract from the flow of the description, but the opposite is true here, integrating the details into the text would've made it more concrete in a useful way and the tables of parameters, as tables, would not have interrupted the flow while making it much easier to see the scale of the models and their architecture.

*Reviewer #3 (Recommendations for the authors):*

The paper by Xie et al. is a modelling study of the mossy fiber-to-granule cell-to-Purkinje cell network, reporting that the optimal type of representations in the cerebellar granule cell layer depends on the type task. The paper stresses that the findings indicate a higher overall bias towards dense representations than stated in the literature, but it appears the authors have missed parts of the literature that already reported on this. While the modelling and analysis appear mathematically solid, the model is lacking many known constraints of the cerebellar circuitry, which makes the applicability of the findings to the biological counterpart somewhat limited.

I have some concerns with the novelty of the main conclusion, here from the abstract:

'Here, we generalize theories of cerebellar learning to determine the optimal granule cell representation for tasks beyond random stimulus discrimination, including continuous input-output transformations as required for smooth motor control. We show that for such tasks, the optimal granule cell representation is substantially denser than predicted by classic theories.'

Stated like this, this has in principle already been shown, i.e. for example:

Spanne and Jorntell (2013) Processing of multi-dimensional sensorimotor information in the spinal and cerebellar neuronal circuitry: a new hypothesis. PLoS Comput Biol. 9(3):e1002979.

Indeed, even the 2 DoF arm movement control that is used in the present paper as an application, was used in this previous paper, with similar conclusions with respect to the advantage of continuous input-output transformations and dense coding. Thus, already from the beginning of this paper, the novelty aspect of this paper is questionable. Even the conclusion in the last paragraph of the Introduction: 'We show that, when learning input-output mappings for motor control tasks, the optimal granule cell representation is much denser than predicted by previous analyses.' was in principle already shown by this previous paper.

However, the present paper does add several more specific investigations/characterizations that were not previously explored. Many of the main figures report interesting new model results. However, the model is implemented in a highly generic fashion. Consequently, the model relates better to general neural network theory than to specific interpretations of the function of the cerebellar neuronal circuitry. One good example is the findings reported in Figure 2. These represent an interesting extension to the main conclusion, but they are also partly based on arbitrariness as the type of mossy fiber input described in the random categorization task has not been observed in the mammalian cerebellum under behavior in vivo, whereas in contrast, the type of input for the motor control task does resemble mossy fiber input recorded under behavior (van Kan et al. 1993).

The overall conclusion states:

'Our results…suggest that optimal cerebellar representations are task-dependent.'

This is not a particularly strong or specific conclusion. One could interpret this statement as simply saying: ' if I construct an arbitrary neural network, with arbitrary intrinsic properties in neurons and synapses, I can get outputs that depend on the intensity of the input that I provide to that network.'

Further, the last sentence of the Introduction states: 'More broadly, we show that the sparsity of a neural code has a task-dependent inﬂuence on learning…' This is very general and unspecific, and would likely not come as a surprise to anyone interested in the analysis of neural networks. It doesn't pinpoint any specific biological problem but just says that if I change the density of the input to a [generic] network, then the learning will be impacted in one way or another.

The interpretation of the distribution of the mossy fiber inputs to the granule cells, which would have a crucial impact on the results of a study like this, is likely incorrect. First, unlike the papers that the authors cite, there are many studies indicating that there is a topographic organization in the mossy fiber termination, such that mossy fibers from the same inputs, representing similar types of information, are regionally co-localized in the granule cell layer. Hence, there is no support for the model assumption that there is a predominantly random termination of mossy fibers of different origins. This risks invalidating the comparisons that the authors are making, i.e. such as in Figure 3. This is a list of example papers, there are more:

van Kan, Gibson and Houk (1993) Movement-related inputs to intermediate cerebellum of the monkey. Journal of Neurophysiology.

Garwicz et al. (1998) Cutaneous receptive fields and topography of mossy fibres and climbing fibres projecting to cat cerebellar C3 zone. The Journal of Physiology.

Brown and Bower (2001) Congruence of mossy fiber and climbing fiber tactile projections in the lateral hemispheres of the rat cerebellum. The Journal of Comparative Neurology.

Na, Sugihara, Shinoda (2019) The entire trajectories of single pontocerebellar axons and their lobular and longitudinal terminal distribution patterns in multiple aldolase C-positive compartments of the rat cerebellar cortex. The Journal of Comparative Neurology.

The nature of the mossy fiber-granule cell recording is also reviewed here:

Gilbert and Miall (2022) How and Why the Cerebellum Recodes Input Signals: An Alternative to Machine Learning. The Neuroscientist

Further, considering the recoding idea, the following paper shows that detailed information, as it is provided by mossy fibers, is transmitted through the granule cells without any evidence of recoding: Jorntell and Ekerot (2006) Journal of Neuroscience; and this paper shows that these granule inputs are powerfully transmitted to the molecular layer even in a decerebrated animal (i.e. where only the ascending sensory pathways remains) Jorntell and Ekerot 2002, Neuron.

I could not find any description of the neuron model used in this paper, so I assume that the neurons are just modelled as linear summators with a threshold (in fact, Figure 5 mentions inhibition, but this appears to be just one big lump inhibition, which basically is an incorrect implementation). In reality, granule cells of course do have specific properties that can impact the input-output transformation, PARTICULARLY with respect to the comparison of sparse versus dense coding, because the low-pass filtering of input that occurs in granule cells (and other neurons) as well as their spike firing stochasticity (Saarinen et al. (2008). Stochastic differential equation model for cerebellar granule cell excitability. PLoS Comput. Biol. 4:e1000004) will profoundly complicate these comparisons and make them less straight forward than what is portrayed in this paper. There are also several other factors that would be present in the biological setting but are lacking here, which makes it doubtful how much information in relation to the biological performance that this modelling study provides:

What are the types of activity patterns of the inputs? What are the learning rules? What is the topography? What is the impact of Purkinje cell outputs downstream, as the Purkinje cell output does not have any direct action, it acts on the deep cerebellar nuclear neurons, which in turn act on a complex sensorimotor circuitry to exert their effect, hence predictive coding could only become interpretable after the PC output has been added to the activity in those circuits. Where is the differentiated Golgi cell inhibition?

The problem of these, in my impression, generic, arbitrary settings of the neurons and the network in the model becomes obvious here: 'In contrast to the dense activity in cerebellar granule cells, odor responses in Kenyon cells, the analogs of granule cells in the *Drosophila* mushroom body, are sparse…' How can this system be interpreted as an analogy to granule cells in the mammalian cerebellum when the model does not address the specifics lined up above? I.e. the 'inductive bias' that the authors speak of, defined as 'the tendency of a network toward learning particular types of input-output mappings', would be highly dependent on the specifics of the network model.

More detailed comments:

Abstract:

'In these models [Marr-Albus], granule cells form a sparse, combinatorial encoding of diverse sensorimotor inputs. Such sparse representations are optimal for learning to discriminate random stimuli.' Yes, I would agree with the first part, but I contest the second part of this statement. I think what is true for sparse coding is that the learning of random stimuli will be faster, as in a perceptron, but not necessarily better. As the sparsification essentially removes information, it could be argued that the quality of the learning is poorer. So from that perspective, it is not optimal. The authors need to specify from what perspective they consider sparse representations optimal for learning.

Introduction:

'Indeed, several recent studies have reported dense activity in cerebellar granule cells in response to sensory stimulation or during motor control tasks (Knogler et al., 2017; Wagner et al., 2017; Giovannucci et al., 2017; Badura and De Zeeuw, 2017; Wagner et al., 2019), at odds with classic theories (Marr, 1969; Albus, 1971).' In fact, this was precisely the issue that was addressed already by Jorntell and Ekerot (2006) Journal of Neuroscience. The conclusion was that these actual recordings of granule cells in vivo provided essentially no support for the assumptions in the Marr-Albus theories.

Results:

First para: There is no information about how the granule cells are modelled.

Second para: 'A typical assumption in computational theories of the cerebellar cortex is that inputs are randomly distributed in a high-dimensional space.' Yes, I agree, and this is in fact in conflict with the known topographical organization in the cerebellar cortex (see broader comment above). Mossy fiber inputs coding for closely related inputs are co-localized in the cerebellar cortex. I think for this model to be of interest from the point of view of the mammalian cerebellar cortex, it would need to pay more attention to this organizational feature.

---

## [Author Response]

Essential revisions:1) All reviewers had questions about specific details of the study (see below comments). I hope you can use these questions as a guideline to improve the clarity and accessibility of the paper. As suggested by reviewer #2, this may involve moving some details from the supplementary materials to the main manuscript.

As detailed in our response to Reviewer 2, we have moved a number of equations that clarify our model to the main text, as well as including Table 1 which provides details about model parameters. Furthermore, we have substantially extended our Discussion section on Assumptions and Limitations, which addresses comments on biological plausibility.

2) The current eLife assessment (see below) emphasizes the main limitations of the paper – namely the question of to what degree the conclusions of the paper would change if more specific (and biologically more plausible) details about the cerebellar circuit would be taken into account. I hope you will take the opportunity to respond in detail to these points in your response to the reviewer document, which will be posted alongside the revised version of the manuscript and cover the main issues in the Discussion section of the main document.

As mentioned above, the Assumptions and Limitations section section now includes an extended discussion of several issues relating to biological plausibility, including randomness in connectivity onto granule cells, topographic organization of the cerebellar cortex, influence of Golgi cell inhibition, and several other topics. We have additionally added a figure supplement (Figure 7—figure supplement 2) showing that our qualitative results hold when learning is mediated by an online climbing fiber-dependent learning rule, under the assumption that climbing fibers encode an error signal.

Reviewer #1 (Recommendations for the authors):This paper provides compelling and clear analyses that show that the coding level (sparsity) of the granule-cell layer of a cerebellar-like network does not only change the dimensionality of the representation, but also the inductive bias of the network: Higher sparsity biases the network to learning more high-frequency representations. Depending on the dominant frequencies of the target function, different coding levels are therefore optimal. The results are important/fundamental to theories of cerebellar learning and speak to a relevant ongoing debate in cerebellar learning, but will be of interest to readers outside of cerebellar neurophysiology.

We appreciate the Reviewer’s positive comments.

I had two problems in understanding the paper that the authors hopefully can clarify in the revision:Page 8: Third paragraph: At this point in the text it is a bit unclear why the K(x * x') changes shape as shown in Figure 3c. I assume the shape of the kernel K depends on the activation function in the hidden layer. It may be useful for the reader if you could give an example of the Kernel for the specific case you are showing in Figure 3c.

We agree this point needed more clarity. Figure 3C shows the kernel for a rectified linear (ReLU) activation function. The key qualitative effect is the dependence of the kernel on the coding level *f*, and this effect is present across different choices of activation function (see Figure 2—figure supplement 2). We show the influence of coding level on the kernel visually in Figures 3a and 3b and in Equations 1 and 2. Equation 2 shows that the kernel depends on the hidden layer activation h(x), and Equation 1 shows h(x) depends on the threshold *θ*.

To communicate this idea more clearly, we added the following sentences (see bolded text) to Figure 3C’s caption:

(“C) Kernel *K*(x*,*x^0^) for networks with rectified linear activation functions (Equation 1), normalized so that fully overlapping representations have an overlap of 1, plotted as a function of overlap in the space of task variables. The vertical axis corresponds to the ratio of the area of the purple region to the area of the red or blue regions in (B). Each curve corresponds to the kernel of an infinite-width network with a different coding level *f*.”

In the main text, in the last paragraph on page 8, we also added the following text:

Equations 1 and 2 show that the threshold *θ*, which determines the coding level, influences the kernel through its effect on the expansion layer activity h(x).

Page 11, 4th paragraph: Why is the distribution corr(J^eff^_i_,J^eff^_j_) uniform on -1 to 1? Since both are high-dimensional random vectors, should the correlation not be centered around 0? The target of uniform distribution needs to be better explained to make this analysis accessible.

We agree that the uniform distribution might be unexpected. This is in fact a consequence of our assumption in this figure that the points are drawn from a task subspace of dimension *D* = 3. For a unit sphere in *D* = 3 dimensions, the dot product between randomly chosen pairs of points is uniformly distributed on [−1*,*1]. For higher dimensions, the distribution, as the Reviewer expects, is centered at zero but decays away from zero (it is a type of β distribution). We have added a sentence to the figure caption stating this:

(“C) Distributions of synaptic weight correlations Corr  (jieff,jjeff), where J^eff^*_i_* is the *i*th row of J^eff^, for pairs of expansion layer neurons in networks with different numbers of input layer neurons *N* (colors) when *K* = 4 and *D* = 3. Black distribution corresponds to fully connected networks with Gaussian weights. We note that when *D* = 3, the distribution of correlations for random Gaussian weight vectors is uniform on [−1*,*1] as shown (for higher dimensions the distribution has a peak at 0).”*Reviewer #2 (Recommendations for the authors):*

Here, a simple model of cerebellar computation is used to study the dependence of task performance on input type: it is demonstrated that task performance and optimal representations are highly dependent on task and stimulus type. This challenges many standard models which use simple random stimuli and concludes that the granular layer is required to provide a sparse representation. This is a useful contribution to our understanding of cerebellar circuits, though, in common with many models of this type, the neural dynamics and circuit architecture are not very specific to the cerebellum, the model includes the feed-forward structure and the high dimension of the granule layer, but little else. This paper has the virtue of including tasks that are more realistic, but by the paper's own admission, the same model can be applied to the electrosensory lateral line lobe and it could, though it is not mentioned in the paper, be applied to the dentate gyrus and large pyramidal cells of CA3. The discussion does not include specific elements related to, for example, the dynamics of the Purkinje cells or the role of Golgi cells, and, in a way, the demonstration that the model can encompass different tasks and stimuli types is an indication of how abstract the model is. Nonetheless, it is useful and interesting to see a generalization of what has become a standard paradigm for discussing cerebellar function.

We appreciate the Reviewer’s positive comments. Regarding the simplifications of our model, we agree that we have taken a modeling approach that abstracts away certain details to permit comparisons across systems. We now include an in-depth discussion of our simplifying assumptions (Assumptions and Extensions section in the Discussion) and have further noted the possibility that other biophysical mechanisms we have not accounted for may also underlie differences across systems.

Our results predict that qualitative differences in the coding levels of cerebellum-like systems, across brain regions or across species, reflect an optimization to distinct tasks (Figure 7). However, it is also possible that differences in coding level arise from other physiological differences between systems.

I was impressed by the clarity of this manuscript. My only comment is that I found too much was deferred to the appendix, I thought the bits and pieces in the appendix were very clarifying, and given the appendix contains short pieces of extra information explaining the exact nature of the models and tasks, the manuscript would have been easier to follow and think about if these had just been integrated into the text. Often you read papers and wish some details had been shifted into an appendix since they distract from the flow of the description, but the opposite is true here, integrating the details into the text would've made it more concrete in a useful way and the tables of parameters, as tables, would not have interrupted the flow while making it much easier to see the scale of the models and their architecture.

1) When we introduce the model in the Results section, we clarify that we use ReLU activation functions throughout the study.

The activity of neurons in the expansion layer is given by:

h = *φ*(J^eff^x − *θ*)*,*

where *φ* is a rectified linear activation function *φ*(*u*) = max(*u,*0) applied element-wise. Our results also hold for other threshold-polynomial activation functions.

2) We provide more specifics about the random categorization task for Figure 2:

The former we refer to as a “random categorization task” and is parameterized by the number of input pattern-to-category associations *P* learned during training (Figure 2C). During the training phase, the network learns to associate random input patterns x*^µ^* ∈ R*^D^* for *µ* = 1*,…,P* with random binary categories *y^µ^* = ±1. The elements of x*^µ^* are drawn i.i.d. from a normal distribution with mean 0 and variance 1*/D*. We refer to x*^µ^* as “training patterns.” To assess the network’s generalization performance, it is presented with “test patterns” generated by adding noise (parameterized by a noise magnitude; see Methods) to the training patterns. Tasks with continuous outputs (Figure 2D) are parameterized by a length scale that determines how quickly the output changes as a function of the input (specifically, input-output functions are drawn from a Gaussian process with length scale *γ* for variations in *f*(x) as a function of x; see Methods). In this case, both training and test patterns are drawn uniformly on the unit sphere. Later, we will also consider tasks implemented by specific cerebellum-like systems. See Table 1 for a summary of parameters throughout this study.

3) We specify the readouts and the performance metrics we use:

We trained the readout to approximate the target output for training patterns and generalize to unseen test patterns. The network’s prediction is *f*^ˆ^(x) = w·h(x) for tasks with continuous outputs, or *f*^ˆ^(x) = sign(w·h(x)) for categorization tasks, where w are the synaptic weights of the readout from the expansion layer. These weights were set using least squares regression. Performance was measured as the fraction of incorrect predictions for categorization tasks, or relative mean squared error for tasks with continuous targets: Error = E[(f(x)−f(x))2]E[f(x)2], where the expectation is across test patterns.

4) To clarify differences in model and task parameters used across the different figures, we have moved the table of parameters to the main text. The main text references this table after introducing the model and the tasks used in Figure 2.

Reviewer #3 (Recommendations for the authors):The paper by Xie et al. is a modelling study of the mossy fiber-to-granule cell-to-Purkinje cell network, reporting that the optimal type of representations in the cerebellar granule cell layer depends on the type task. The paper stresses that the findings indicate a higher overall bias towards dense representations than stated in the literature, but it appears the authors have missed parts of the literature that already reported on this. While the modelling and analysis appear mathematically solid, the model is lacking many known constraints of the cerebellar circuitry, which makes the applicability of the findings to the biological counterpart somewhat limited.

We thank the Reviewer for suggesting additional references to include in our manuscript, and for encouraging us to extend our model toward greater biological plausibility and more critically discuss simplifying assumptions we have made. We respond to both the comment about previous literature and about applicability to cerebellar circuitry in detail below.

I have some concerns with the novelty of the main conclusion, here from the abstract:'Here, we generalize theories of cerebellar learning to determine the optimal granule cell representation for tasks beyond random stimulus discrimination, including continuous input-output transformations as required for smooth motor control. We show that for such tasks, the optimal granule cell representation is substantially denser than predicted by classic theories.'Stated like this, this has in principle already been shown, i.e. for example:Spanne and Jorntell (2013) Processing of multi-dimensional sensorimotor information in the spinal and cerebellar neuronal circuitry: a new hypothesis. PLoS Comput Biol. 9(3):e1002979.Indeed, even the 2 DoF arm movement control that is used in the present paper as an application, was used in this previous paper, with similar conclusions with respect to the advantage of continuous input-output transformations and dense coding. Thus, already from the beginning of this paper, the novelty aspect of this paper is questionable. Even the conclusion in the last paragraph of the Introduction: 'We show that, when learning input-output mappings for motor control tasks, the optimal granule cell representation is much denser than predicted by previous analyses.' was in principle already shown by this previous paper.

We thank the Reviewer for drawing our attention to Spanne and Jo¨rntell (2013). Our study shares certain similarities with this work, including the consideration of tasks with smooth input-output mappings, such as learning the dynamics of a two-joint arm. However, our study differs substantially, most notably the fact that we focus our study on parametrically varying the degree of sparsity in the granule cell layer to determine the circumstances under which dense versus sparse coding is optimal. To the best of our ability, we can find no result in Spanne and J¨orntell (2013) that indicates the performance of a network as a function of average coding level. Instead, Spanne and Jo¨rntell (2013) propose that inhibition from Golgi cells produces heterogeneity in coding level which can improve performance, which is an interesting but complementary finding to ours. We therefore do not believe that the quantitative computations of optimal coding level that we present are redundant with the results of this previous study. We also note that a key contribution of our study is mathemetical analysis of the inductive bias of networks with different coding levels which supports our conclusions.

We have included a discussion of Spanne and Jo¨rntell (2013) and (2015) in the revised version of our manuscript:

“Other studies have considered tasks with smooth input-output mappings and low-dimensional inputs, finding that heterogeneous Golgi cell inhibition can improve performance by diversifying individual granule cell thresholds (Spanne and J¨orntell, 2013). Extending our model to include heterogeneous thresholds is an interesting direction for future work. Another proposal states that dense coding may improve generalization (Spanne and Jo¨rntell, 2015). Our theory reveals that whether or not dense coding is beneficial depends on the task.”

However, the present paper does add several more specific investigations/characterizations that were not previously explored. Many of the main figures report interesting new model results. However, the model is implemented in a highly generic fashion. Consequently, the model relates better to general neural network theory than to specific interpretations of the function of the cerebellar neuronal circuitry. One good example is the findings reported in Figure 2. These represent an interesting extension to the main conclusion, but they are also partly based on arbitrariness as the type of mossy fiber input described in the random categorization task has not been observed in the mammalian cerebellum under behavior in vivo, whereas in contrast, the type of input for the motor control task does resemble mossy fiber input recorded under behavior (van Kan et al. 1993).

We agree that the tasks we consider in Figure 2 are simplified compared to those that we consider elsewhere in the paper. The choice of random mossy fiber input was made to provide a comparison to previous modeling studies that also use random input as a benchmark (Marr 1969, Albus 1971, Brunel 2004, Babadi and Sompolinsky 2014, Billings 2014, LitwinKumar et al., 2017). This baseline permits us to specifically evaluate the effects of lowdimensional inputs (Figure 2) and richer input-output mappings (Figure 2, Figure 7). We agree with the Reviewer that the random and uncorrelated mossy fiber activity that has been extensively used in previous studies is almost certainly an unrealistic idealization of in vivo neural activity—this is a motivating factor for our study, which relaxes this assumption and examines the consequences. To provide additional context, we have updated the following paragraph in the main text Results section:

“A typical assumption in computational theories of the cerebellar cortex is that inputs are randomly distributed in a high-dimensional space (Marr, 1969; Albus, 1971; Brunel et al., 2004; Babadi and Sompolinsky, 2014; Billings et al., 2014; Litwin-Kumar et al., 2017). While this may be a reasonable simplification in some cases, many tasks, including cerebellumdependent tasks, are likely best-described as being encoded by a low-dimensional set of variables. For example, the cerebellum is often hypothesized to learn a forward model for motor control (Wolpert et al., 1998), which uses sensory input and motor efference to predict an effector’s future state. Mossy fiber activity recorded in monkeys correlates with position and velocity during natural movement (van Kan et al., 1993). Sources of motor efference copies include motor cortex, whose population activity lies on a lowdimensional manifold (Wagner et al., 2019; Huang et al., 2013; Churchland et al., 2010; Yu et al., 2009). We begin by modeling the low dimensionality of inputs and later consider more specific tasks.”

The overall conclusion states:'Our results….suggest that optimal cerebellar representations are task-dependent.'This is not a particularly strong or specific conclusion. One could interpret this statement as simply saying: ' if I construct an arbitrary neural network, with arbitrary intrinsic properties in neurons and synapses, I can get outputs that depend on the intensity of the input that I provide to that network.'Further, the last sentence of the Introduction states: 'More broadly, we show that the sparsity of a neural code has a task-dependent inﬂuence on learning…' This is very general and unspecific, and would likely not come as a surprise to anyone interested in the analysis of neural networks. It doesn't pinpoint any specific biological problem but just says that if I change the density of the input to a [generic] network, then the learning will be impacted in one way or another.

We agree with the Reviewer that our conclusions are quite general, and we have removed the final sentence as we agree it was unspecific. However, we disagree with the Reviewer’s paraphrasing of our results.

First, we do not select arbitrary intrinsic properties of neurons and synapses. Rather, we construct a simplified model with a key quantity, the neuronal threshold, that we vary parametrically in order to assess the effect of the resulting changes in the representation on performance. Second, we do not vary the intensity/density of inputs provided to the network – this is fixed throughout our study for all key comparisons we perform. Instead, we vary the density (coding level) of the expansion layer representation and quantify its effect on inductive bias and generalization. Finally, our study’s key contribution is an explanation of the heterogeneity in average coding level observed across behaviors and cerebellum-like systems. We go beyond the empirical statement that there is a dependence of performance on the parameter that we vary by developing an analytical theory. Our theory describes the performance of the class of networks that we study and the properties of learning tasks that determine the optimal expansion layer representation.

To clarify our main contributions, we have updated the final paragraph of the Introduction. We have also removed the sentence that the Reviewer objects to, as it was less specific than the other points we make here.

We propose that these differences can be explained by the capacity of representations with different levels of sparsity to support learning of different tasks. We show that the optimal level of sparsity depends on the structure of the input-output relationship of a task. When learning input-output mappings for motor control tasks, the optimal granule cell representation is much denser than predicted by previous analyses. To explain this result, we develop an analytic theory that predicts the performance of cerebellum-like circuits for arbitrary learning tasks. The theory describes how properties of cerebellar architecture and activity control these networks’ inductive bias: the tendency of a network toward learning particular types of input-output mappings (Sollich, 1998; Jacot et al., 2018; Bordelon et al., 2020; Canatar et al., 2021; Simon et al., 2021). The theory shows that inductive bias, rather than the dimension of the representation alone, is necessary to explain learning performance across tasks. It also suggests that cerebellar regions specialized for different functions may adjust the sparsity of their granule cell representations depending on the task.

The interpretation of the distribution of the mossy fiber inputs to the granule cells, which would have a crucial impact on the results of a study like this, is likely incorrect. First, unlike the papers that the authors cite, there are many studies indicating that there is a topographic organization in the mossy fiber termination, such that mossy fibers from the same inputs, representing similar types of information, are regionally co-localized in the granule cell layer. Hence, there is no support for the model assumption that there is a predominantly random termination of mossy fibers of different origins. This risks invalidating the comparisons that the authors are making, i.e. such as in Figure 3. This is a list of example papers, there are more:van Kan, Gibson and Houk (1993) Movement-related inputs to intermediate cerebellum of the monkey. Journal of Neurophysiology.Garwicz et al. (1998) Cutaneous receptive fields and topography of mossy fibres and climbing fibres projecting to cat cerebellar C3 zone. The Journal of Physiology.Brown and Bower (2001) Congruence of mossy fiber and climbing fiber tactile projections in the lateral hemispheres of the rat cerebellum. The Journal of Comparative Neurology.Na, Sugihara, Shinoda (2019) The entire trajectories of single pontocerebellar axons and their lobular and longitudinal terminal distribution patterns in multiple aldolase C-positive compartments of the rat cerebellar cortex. The Journal of Comparative Neurology.The nature of the mossy fiber-granule cell recording is also reviewed here:Gilbert and Miall (2022) How and Why the Cerebellum Recodes Input Signals: An Alternative to Machine Learning. The NeuroscientistFurther, considering the recoding idea, the following paper shows that detailed information, as it is provided by mossy fibers, is transmitted through the granule cells without any evidence of recoding: Jorntell and Ekerot (2006) Journal of Neuroscience; and this paper shows that these granule inputs are powerfully transmitted to the molecular layer even in a decerebrated animal (i.e. where only the ascending sensory pathways remains) Jorntell and Ekerot 2002, Neuron.

We agree that there is strong evidence for a topographic organization in mossy fiber to granule cell connectivity at the microzonal level. We thank the Reviewer for pointing us to specific examples. We acknowledge that our simplified model does not capture the structure of connectivity observed in these studies.

However, the focus of our model is on cerebellar neurons presynaptic to a single Purkinje cell. Random or disordered distribution of inputs at this local scale is compatible with topographic organization at the microzonal scale. Furthermore, while there is evidence of structured connections at the local scale, models with random connectivity are able to reproduce the dimensionality of granule cell activity within a small margin of error (Nguyen et al., 2022). Finally, our finding that dense codes are optimal for learning slowly varying tasks is consistent with evidence for the lack of re-coding – for such tasks, re-coding may absent because it is not required.

We have dedicated a section on this issue in the Assumptions and Extensions portion of our Discussion:

“Another key assumption concerning the granule cells is that they sample mossy fiber inputs randomly, as is typically assumed in Marr-Albus models (Marr, 1969; Albus, 1971; LitwinKumar et al., 2017; Cayco-Gajic et al., 2017). Other studies instead argue that granule cells sample from mossy fibers with highly similar receptive fields (Garwicz et al., 1998; Brown and Bower, 2001; J¨orntell and Ekerot, 2006) defined by the tuning of mossy fiber and climbing fiber inputs to cerebellar microzones (Apps et al., 2018). This has led to an alternative hypothesis that granule cells serve to relay similarly tuned mossy fiber inputs and enhance their signal-to-noise ratio (Jo¨rntell and Ekerot, 2006; Gilbert and Chris Miall, 2022) rather than to re-encode inputs. Another hypothesis is that granule cells enable Purkinje cells to learn piece-wise linear approximations of nonlinear functions (Spanne and J¨orntell, 2013). However, several recent studies support the existence of heterogeneous connectivity and selectivity of granule cells to multiple distinct inputs at the local scale (Huang et al., 2013; Ishikawa et al., 2015). Furthermore, the deviation of the predicted dimension in models constrained by electron-microscopy data as compared to randomly wired models is modest (Nguyen et al., 2022). Thus, topographically organized connectivity at the macroscopic scale may coexist with disordered connectivity at the local scale, allowing granule cells presynaptic to an individual Purkinje cell to sample heterogeneous combinations of the subset of sensorimotor signals relevant to the tasks that Purkinje cell participates in. Finally, we note that the optimality of dense codes for learning slowly varying tasks in our theory suggests that observations of a lack of mixing (J¨orntell and Ekerot, 2002) for such tasks are compatible with Marr-Albus models, as in this case nonlinear mixing is not required.”

I could not find any description of the neuron model used in this paper, so I assume that the neurons are just modelled as linear summators with a threshold (in fact, Figure 5 mentions inhibition, but this appears to be just one big lump inhibition, which basically is an incorrect implementation). In reality, granule cells of course do have specific properties that can impact the input-output transformation, PARTICULARLY with respect to the comparison of sparse versus dense coding, because the low-pass filtering of input that occurs in granule cells (and other neurons) as well as their spike firing stochasticity (Saarinen et al. (2008). Stochastic differential equation model for cerebellar granule cell excitability. PLoS Comput. Biol. 4:e1000004) will profoundly complicate these comparisons and make them less straight forward than what is portrayed in this paper. There are also several other factors that would be present in the biological setting but are lacking here, which makes it doubtful how much information in relation to the biological performance that this modelling study provides:What are the types of activity patterns of the inputs? What are the learning rules? What is the topography? What is the impact of Purkinje cell outputs downstream, as the Purkinje cell output does not have any direct action, it acts on the deep cerebellar nuclear neurons, which in turn act on a complex sensorimotor circuitry to exert their effect, hence predictive coding could only become interpretable after the PC output has been added to the activity in those circuits. Where is the differentiated Golgi cell inhibition?

Thank you for these critiques. We have made numerous edits to improve the presentation of the details of our model in the main text of the manuscript. Indeed, granule cells in the main text are modeled as linear sums of mossy fiber inputs with a threshold-linear activation function. A more detailed description of the model for granule cells can now be found in Equation 1 in the Results section:

“The activity of neurons in the expansion layer is given by:

h = *φ*(J^eff^x − *θ*)*,* (1)

where *φ* is a rectified linear activation function *φ*(*u*) = max(*u,*0) applied element-wise. Our results also hold for other threshold-polynomial activation functions. The scalar threshold *θ* is shared across neurons and controls the coding level, which we denote by *f*, defined as the average fraction of neurons in the expansion layer that are active.”

Most of our analyses use the firing rate model we describe above, but several Supplemental Figures show extensions to this model. As we mention in the Discussion, our results do not depend on the specific choice of nonlinearity (Figure 2—figure supplement 2). We have also considered the possibility that the stochastic nature of granule cell spikes could impact our measures of coding level. In Figure 7—figure supplement 1 we test the robustness of our main conclusion using a spiking model where we model granule cell spikes with Poisson statistics. When measuring coding level in a population of spiking neurons, a key question is at what time window the Purkinje cell integrates spikes. For several choices of integration time windows, we show that dense coding remains optimal for learning smooth tasks. However, we agree with the Reviewer that there are other biological details our model does not address. For example, our spiking model does not capture some of the properties the Saarinen et al. (2008) model captures, including random sub-threshold oscillations and clusters of spikes. Modeling biophysical phenomena at this scale is beyond the scope of our study. We have added this reference to the relevant section of the Discussion:

“We also note that coding level is most easily defined when neurons are modeled as rate, rather than spiking units. To investigate the consistency of our results under a spiking code, we implemented a model in which granule cell spiking exhibits Poisson variability and quantify coding level as the fraction of neurons that have nonzero spike counts (Figure 7—figure supplement 1; Figure 7C). In general, increased spike count leads to improved performance as noise associated with spiking variability is reduced. Granule cells have been shown to exhibit reliable burst responses to mossy fiber stimulation (Chadderton et al., 2004), motivating models using deterministic responses or sub-Poisson spiking variability. However, further work is needed to quantitatively compare variability in model and experiment and to account for more complex biophysical properties of granule cells (Saarinen et al., 2008).”

A second concern the Reviewer raises is our implementation of Golgi cell inhibition as a homogeneous rather than heterogeneous input onto granule cells. In simplified models, adding heterogeneous inhibition does not dramatically change the qualitative properties of the expansion layer representation, in particular the dimensionality of the representation (Billings et al., 2014, Cayco-Gajic et al., 2017, Litwin-Kumar et al., 2017). We have added a section about inhibition to our Discussion:

“We also have not explicitly modeled inhibitory input provided by Golgi cells, instead assuming such input can be modeled as a change in effective threshold, as in previous studies (Billings et al., 2014; Cayco-Gajic et al., 2017; Litwin-Kumar et al., 2017). This is appropriate when considering the dimension of the granule cell representation (Litwin-Kumar et al., 2017), but more work is needed to extend our model to the case of heterogeneous inhibition.”

Regarding the mossy fiber inputs, as we state in response to paragraph 3, we agree with the Reviewer that the random and uncorrelated mossy fiber activity that has been used in previous studies is an unrealistic idealization of in vivo neural activity. One of the motivations for our model was to relax this assumption and examine the consequences: we introduce correlations in the mossy fiber activity by projecting low-dimensional patterns into the mossy fiber layer (Figure 1B):

“A typical assumption in computational theories of the cerebellar cortex is that inputs are randomly distributed in a high-dimensional space (Marr, 1969; Albus, 1971; Brunel et al., 2004; Babadi and Sompolinsky, 2014; Billings et al., 2014; Litwin-Kumar et al., 2017). While this may be a reasonable simplification in some cases, many tasks, including cerebellumdependent tasks, are likely best-described as being encoded by a low-dimensional set of variables. For example, the cerebellum is often hypothesized to learn a forward model for motor control (Wolpert et al., 1998), which uses sensory input and motor efference to predict an effector’s future state. Mossy fiber activity recorded in monkeys correlates with position and velocity during natural movement (van Kan et al., 1993). Sources of motor efference copies include motor cortex, whose population activity lies on a low-dimensional manifold (Wagner et al., 2019; Huang et al., 2013; Churchland et al., 2010; Yu et al., 2009). We begin by modeling the low dimensionality of inputs and later consider more specific tasks.

We therefore assume that the inputs to our model lie on a *D*-dimensional subspace embedded in the *N*-dimensional input space, where *D* is typically much smaller than *N* (Figure 1B). We refer to this subspace as the “task subspace” (Figure 1C).”

The Reviewer also mentions the learning rule at granule cell to Purkinje cell synapses. We agree that considering online, climbing-fiber-dependent learning is an important generalization. We therefore added a new supplemental figure investigating whether we would still see a difference in optimal coding levels across tasks if online learning were used instead of the least squares solution (Figure 7—figure supplement 2). Indeed, we observed a similar task dependence as we saw in Figure 2F. We have added a new paragraph in the Discussion under Assumptions and Extensions describing our rationale and approach in detail:

“For the Purkinje cells, our model assumes that their responses to granule cell input can be modeled as an optimal linear readout. Our model therefore provides an upper bound to linear readout performance, a standard benchmark for the quality of a neural representation that does not require assumptions on the nature of climbing fiber-mediated plasticity, which is still debated. Electrophysiological studies have argued in favor of a linear approximation (Brunel et al., 2004). To improve the biological applicability of our model, we implemented an online climbing fiber-mediated learning rule and found that optimal coding levels are still task-dependent (Figure 7—figure supplement 2). We also note that although we model several timing-dependent tasks (Figure 7), our learning rule does not exploit temporal information, and we assume that temporal dynamics of granule cell responses are largely inherited from mossy fibers. Integrating temporal information into our model is an interesting direction for future investigation.

During each epoch of training, the network is presented with all patterns in a randomized order, and the learned weights are updated with each pattern (see Methods). Networks were presented with 30 patterns and trained for 20*,*000 epochs, with a learning rate of *η* = 0*.*7*/M*. Other parameters: *D* = 3*,M* = 10*,*000.

A) Performance of an example network during online learning, measured as relative mean squared error across training epochs. Parameters: *f* = 0*.*3, *γ* = 1.

B) Generalization error as a function of coding level for networks trained with online learning (solid lines) or unregularized least squares (dashed lines) for Gaussian process tasks with different length scales (colors). Standard error of the mean was computed across 20 realizations.”

Finally, regarding the function of the Purkinje cell, our model defines a learning task as a mapping from inputs to target activity in the Purkinje cell and is thus agnostic to the cell’s downstream effects. We clarify this point when introducing the definition of a learning task:

“In our model, a learning task is defined by a mapping from task variables **x** to an output *f*(**x**), representing a target change in activity of a readout neuron, for example a Purkinje cell. The limited scope of this definition implies our results should not strongly depend on the influence of the readout neuron on downstream circuits.”

The problem of these, in my impression, generic, arbitrary settings of the neurons and the network in the model becomes obvious here: 'In contrast to the dense activity in cerebellar granule cells, odor responses in Kenyon cells, the analogs of granule cells in the *Drosophila* mushroom body, are sparse…' How can this system be interpreted as an analogy to granule cells in the mammalian cerebellum when the model does not address the specifics lined up above? I.e. the 'inductive bias' that the authors speak of, defined as 'the tendency of a network toward learning particular types of input-output mappings', would be highly dependent on the specifics of the network model.

We agree with the Reviewer that our model makes several simplifying assumptions for mathematical tractability. However, we note that our study is not the first to draw analogies between cerebellum-like systems, including the mushroom body (Bell et al., 2008; Farris, 2011). All the systems we study feature a sparsely connected, expanded granule-like layer that sends parallel fiber axons onto densely connected downstream neurons known to exhibit powerful synaptic plasticity, thus motivating the key architectural assumptions of our model. We have constrained anatomical parameters of the model using data as available (Table 1). However, we agree with the Reviewer that when making comparisons across species there is always a possibility that differences are due to physiological mechanisms we have not fully understood or captured with a model. As such, we can only present a hypothesis for these differences. We have modified our Discussion section on this topic to clearly state this.

“Our results predict that qualitative differences in the coding levels of cerebellum-like systems, across brain regions or across species, reflect an optimization to distinct tasks (Figure 7). However, it is also possible that differences in coding level arise from other physiological differences between systems.”

More detailed comments:Abstract:'In these models [Marr-Albus], granule cells form a sparse, combinatorial encoding of diverse sensorimotor inputs. Such sparse representations are optimal for learning to discriminate random stimuli.' Yes, I would agree with the first part, but I contest the second part of this statement. I think what is true for sparse coding is that the learning of random stimuli will be faster, as in a perceptron, but not necessarily better. As the sparsification essentially removes information, it could be argued that the quality of the learning is poorer. So from that perspective, it is not optimal. The authors need to specify from what perspective they consider sparse representations optimal for learning.

This is an important point that we would like to clarify. It is not the case that sparse coding simply speeds up learning. In our study and many related works (Barak et al. 2013; Babadi and Sompolinsky 2014; Litwin-Kumar et al. 2017), learning performance is measured based on the generalization ability of the network – the ability to predict correct labels for previously unseen inputs. As our study and previous studies show, sparse codes are optimal in the sense that they minimize generalization error, independent of any effect on learning speed. To communicate this more effectively, we have added the following sentence to the first paragraph of the Introduction:

“Sparsity affects both learning speed (Cayco-Gajic et al., 2017), and generalization, the ability to predict correct labels for previously unseen inputs (Barak et al., 2013; Babadi and Sompolinsky, 2014; Litwin-Kumar et al., 2017).”

Introduction:'Indeed, several recent studies have reported dense activity in cerebellar granule cells in response to sensory stimulation or during motor control tasks (Knogler et al., 2017; Wagner et al., 2017; Giovannucci et al., 2017; Badura and De Zeeuw, 2017; Wagner et al., 2019), at odds with classic theories (Marr, 1969; Albus, 1971).' In fact, this was precisely the issue that was addressed already by Jorntell and Ekerot (2006) Journal of Neuroscience. The conclusion was that these actual recordings of granule cells in vivo provided essentially no support for the assumptions in the Marr-Albus theories.

In our reading, the main finding of J¨orntell and Ekerot (2006) is that individual granule cells are activated by mossy fibers with overlapping receptive fields driven by a single type of somatosensory input. However, there is also evidence of nonlinear mixed selectivity in granule cells in support of the re-coding hypothesis (Huang et al., 2013; Ishikawa et al., 2015). Jo¨rntell and Ekerot (2006) also suggest that the granule cell layer shares similar topographic organization as mossy fibers, organized into microzones. The existence of topographic organization does not invalidate Marr-Albus theories. As we have suggested earlier, a local combinatorial expansion can coexist with a global topographic organization.

We have described these considerations in the Assumptions and Extensions portion of the Discussion:

“Another key assumption concerning the granule cells is that they sample mossy fiber inputs randomly, as is typically assumed in Marr-Albus models (Marr, 1969; Albus, 1971; LitwinKumar et al., 2017; Cayco-Gajic et al., 2017). Other studies instead argue that granule cells sample from mossy fibers with highly similar receptive fields (Garwicz et al., 1998; Brown and Bower, 2001; J¨orntell and Ekerot, 2006) defined by the tuning of mossy fiber and climbing fiber inputs to cerebellar microzones (Apps et al., 2018). This has led to an alternative hypothesis that granule cells serve to relay similarly tuned mossy fiber inputs and enhance their signal-to-noise ratio (Jo¨rntell and Ekerot, 2006; Gilbert and Chris Miall, 2022) rather than to re-encode inputs. Another hypothesis is that granule cells enable Purkinje cells to learn piece-wise linear approximations of nonlinear functions (Spanne and J¨orntell, 2013). However, several recent studies support the existence of heterogeneous connectivity and selectivity of granule cells to multiple distinct inputs at the local scale (Huang et al., 2013; Ishikawa et al., 2015). Furthermore, the deviation of the predicted dimension in models constrained by electron-microscopy data as compared to randomly wired models is modest (Nguyen et al., 2022). Thus, topographically organized connectivity at the macroscopic scale may coexist with disordered connectivity at the local scale, allowing granule cells presynaptic to an individual Purkinje cell to sample heterogeneous combinations of the subset of sensorimotor signals relevant to the tasks that Purkinje cell participates in. Finally, we note that the optimality of dense codes for learning slowly varying tasks in our theory suggests that observations of a lack of mixing (J¨orntell and Ekerot, 2002) for such tasks are compatible with Marr-Albus models, as in this case nonlinear mixing is not required.”

We have also included the Jo¨rntell and Ekerot (2006) study as a citation in the Introduction:

“Indeed, several recent studies have reported dense activity in cerebellar granule cells in response to sensory stimulation or during motor control tasks (Jo¨rntell and Ekerot, 2006; Knogler et al., 2017; Wagner et al., 2017; Giovannucci et al., 2017; Badura and De Zeeuw, 2017; Wagner et al., 2019), at odds with classic theories (Marr, 1969; Albus, 1971).”

Results:First para: There is no information about how the granule cells are modelled.

We agree that this should information should have been more readily available. We now more completely describe the model in the main text. Our model for granule cells can be found in Equation 1 in the Results section and also the Methods (Network Model):

“The activity of neurons in the expansion layer is given by:

h = *φ*(J^eff^x − *θ*)*,* (2)

where *φ* is a rectified linear activation function *φ*(*u*) = max(*u,*0) applied element-wise. Our results also hold for other threshold-polynomial activation functions. The scalar threshold *θ* is shared across neurons and controls the coding level, which we denote by *f*, defined as the average fraction of neurons in the expansion layer that are active.”

Second para: 'A typical assumption in computational theories of the cerebellar cortex is that inputs are randomly distributed in a high-dimensional space.' Yes, I agree, and this is in fact in conflict with the known topographical organization in the cerebellar cortex (see broader comment above). Mossy fiber inputs coding for closely related inputs are co-localized in the cerebellar cortex. I think for this model to be of interest from the point of view of the mammalian cerebellar cortex, it would need to pay more attention to this organizational feature.

As we discuss in our response to paragraphs 5 and 6, we see the random distribution assumption at the local scale (inputs presynaptic to a single Purkinje cell) as being compatible with topographic organization occurring at the microzone scale. Furthermore, as discussed earlier, we specifically model low-dimensional input as opposed to the random and high-dimensional inputs typically studied in prior models.

“A typical assumption in computational theories of the cerebellar cortex is that inputs are randomly distributed in a high-dimensional space (Marr, 1969; Albus, 1971; Brunel et al., 2004; Babadi and Sompolinsky, 2014; Billings et al., 2014; Litwin-Kumar et al., 2017). While this may be a reasonable simplification in some cases, many tasks, including cerebellum dependent tasks, are likely best-described as being encoded by a low-dimensional set of variables. For example, the cerebellum is often hypothesized to learn a forward model for motor control (Wolpert et al., 1998), which uses sensory input and motor efference to predict an effector’s future state. Mossy fiber activity recorded in monkeys correlates with position and velocity during natural movement (van Kan et al., 1993). Sources of motor efference copies include motor cortex, whose population activity lies on a low-dimensional manifold (Wagner et al., 2019; Huang et al., 2013; Churchland et al., 2010; Yu et al., 2009). We begin by modeling the low dimensionality of inputs and later consider more specific tasks.

We therefore assume that the inputs to our model lie on a *D*-dimensional subspace embedded in the *N*-dimensional input space, where *D* is typically much smaller than *N* (Figure 1B).

We refer to this subspace as the “task subspace” (Figure 1C).”

References

Albus, J.S. (1971). A theory of cerebellar function. Mathematical Biosciences *10*, 25–61.

Apps, R., et al. (2018). Cerebellar Modules and Their Role as Operational Cerebellar Processing Units. Cerebellum *17*, 654–682.

Babadi, B. and Sompolinsky, H. (2014). Sparseness and expansion in sensory representations. Neuron *83*, 1213–1226.

Badura, A. and De Zeeuw, C.I. (2017). Cerebellar granule cells: dense, rich and evolving representations. Current Biology *27*, R415–R418.

Barak, O., Rigotti, M., and Fusi, S. (2013). The sparseness of mixed selectivity neurons controls the generalization–discrimination trade-off. Journal of Neuroscience *33*, 3844– 3856.

Bell, C.C., Han, V., and Sawtell, N.B. (2008). Cerebellum-like structures and their implications for cerebellar function. Annual Review of Neuroscience *31*, 1–24.

Billings, G., Piasini, E., Lo˝rincz, A., Nusser, Z., and Silver, R.A. (2014). Network structure within the cerebellar input layer enables lossless sparse encoding. Neuron *83*, 960–974.

Bordelon, B., Canatar, A., and Pehlevan, C. (2020). Spectrum dependent learning curves in kernel regression and wide neural networks. International Conference on Machine Learning 1024–1034.

Brown, I.E. and Bower, J.M. (2001). Congruence of mossy fiber and climbing fiber tactile projections in the lateral hemispheres of the rat cerebellum. Journal of Comparative Neurology *429*, 59–70.

Brunel, N., Hakim, V., Isope, P., Nadal, J.P., and Barbour, B. (2004). Optimal information storage and the distribution of synaptic weights: perceptron versus Purkinje cell. Neuron *43*, 745–757.

Canatar, A., Bordelon, B., and Pehlevan, C. (2021). Spectral bias and task-model alignment explain generalization in kernel regression and infinitely wide neural networks. Nature Communications *12*, 1–12.

Cayco-Gajic, N.A., Clopath, C., and Silver, R.A. (2017). Sparse synaptic connectivity is required for decorrelation and pattern separation in feedforward networks. Nature Communications *8*, 1–11.

Chadderton, P., Margrie, T.W., and Ha¨usser, M. (2004). Integration of quanta in cerebellar granule cells during sensory processing. Nature *428*, 856–860.

Churchland, M.M., et al. (2010). Stimulus onset quenches neural variability: a widespread cortical phenomenon. Nature Neuroscience *13*, 369–378.

Farris, S.M. (2011). Are mushroom bodies cerebellum-like structures? Arthropod structure and development *40*, 368–379.

Garwicz, M., Jorntell, H., and Ekerot, C.F. (1998). Cutaneous receptive fields and topography of mossy fibres and climbing fibres projecting to cat cerebellar C3 zone. The Journal of Physiology *512 ( Pt 1)*, 277–293.

Gilbert, M. and Chris Miall, R. (2022). How and Why the Cerebellum Recodes Input Signals: An Alternative to Machine Learning. The Neuroscientist *28*, 206–221.

Giovannucci, A., et al. (2017). Cerebellar granule cells acquire a widespread predictive feedback signal during motor learning. Nature Neuroscience *20*, 727–734.

Huang, C.C., et al. (2013). Convergence of pontine and proprioceptive streams onto multimodal cerebellar granule cells. *eLife 2*, e00400.

Ishikawa, T., Shimuta, M., and Ha¨usser, M. (2015). Multimodal sensory integration in single cerebellar granule cells in vivo. *eLife 4*, e12916.

Jacot, A., Gabriel, F., and Hongler, C. (2018). Neural tangent kernel: Convergence and generalization in neural networks. Advances in Neural Information Processing Systems *31*.

Jo¨rntell, H. and Ekerot, C.F. (2002). Reciprocal Bidirectional Plasticity of Parallel Fiber Receptive Fields in Cerebellar Purkinje Cells and Their Afferent Interneurons. Neuron *34*, 797–806.

Jo¨rntell, H. and Ekerot, C.F. (2006). Properties of Somatosensory Synaptic Integration in Cerebellar Granule Cells in vivo. Journal of Neuroscience *26*, 11786–11797.

Knogler, L.D., Markov, D.A., Dragomir, E.I., Stih, V., and Portugues, R. (2017). Senso-ˇ rimotor representations in cerebellar granule cells in larval zebrafish are dense, spatially organized, and non-temporally patterned. Current Biology *27*, 1288–1302.

Litwin-Kumar, A., Harris, K.D., Axel, R., Sompolinsky, H., and Abbott, L.F. (2017). Optimal degrees of synaptic connectivity. Neuron *93*, 1153–1164.

Marr, D. (1969). A theory of cerebellar cortex. Journal of Physiology *202*, 437–470.

Nguyen, T.M., et al. (2022). Structured cerebellar connectivity supports resilient pattern separation. Nature 1–7.

Saarinen, A., Linne, M.L., and Yli-Harja, O. (2008). Stochastic Differential Equation Model for Cerebellar Granule Cell Excitability. PLOS Computational Biology *4*, e1000004.

Simon, J.B., Dickens, M., and DeWeese, M.R. (2021). A theory of the inductive bias and generalization of kernel regression and wide neural networks. arXiv: 2110.03922.

Sollich, P. (1998). Learning curves for Gaussian processes. Advances in Neural Information Processing Systems *11*.

Spanne, A. and Jo¨rntell, H. (2013). Processing of Multi-dimensional Sensorimotor Information in the Spinal and Cerebellar Neuronal Circuitry: A New Hypothesis. PLOS Computational Biology *9*, e1002979.

Spanne, A. and Jo¨rntell, H. (2015). Questioning the role of sparse coding in the brain. Trends in Neurosciences *38*, 417–427.

van Kan, P.L., Gibson, A.R., and Houk, J.C. (1993). Movement-related inputs to intermediate cerebellum of the monkey. Journal of Neurophysiology *69*, 74–94.

Wagner, M.J., Kim, T.H., Savall, J., Schnitzer, M.J., and Luo, L. (2017). Cerebellar granule cells encode the expectation of reward. Nature *544*, 96–100.

Wagner, M.J., et al. (2019). Shared cortex-cerebellum dynamics in the execution and learning of a motor task. Cell *177*, 669–682.e24.

Wolpert, D.M., Miall, R.C., and Kawato, M. (1998). Internal models in the cerebellum. Trends in Cognitive Sciences *2*, 338–347.

Yu, B.M., et al. (2009). Gaussian-process factor analysis for low-dimensional single-trial analysis of neural population activity. Journal of Neurophysiology *102*, 614–635.